# Endogenous tyrosinase-catalyzed therapeutics

Yawen You[1,2,3], Zhaochen Guo [4], Yixin Wang[1,2,3], Sichen Yuan [1,2,3] & Quanyin Hu [1,2,3] ✉

Tyrosinase (TYR) catalyzes the two initial steps of melanin synthesis from tyrosine in various organisms. However, overproduction, accumulation, and abnormal reduction of melanin can lead to severe diseases, particularly skin diseases, which makes tyrosinase a significant endogenous target in developing therapeutics to treat melanin-associated disorders. Herein, we devise a TYR-based in situ catalytic platform that can generate drugs intracellularly through an endogenous copper-catalyzed azide-alkyne cycloaddition (CuAAC) reaction. By taking advantage of the potent catalytic activity of TYR that is mechanistically validated by ab initio molecular dynamics (AIMD) theoretical calculation and experimental catalysis performance, we develop a TYR-catalyzed in-situ formed proteolysis-targeting chimeras (PROTACs) to degrade intracellular TYR protein to decrease melanin synthesis for treating hyperpigmentation and a TYR-catalyzed in-situ activated prodrug strategy to overcome drug resistance for melanoma therapy. In male mouse models, we show that this TYR-catalyzed therapeutics could efficiently alleviate skin hyperpigmentation within 48 h as well as resensitize the drug-resistant melanoma cells to chemotherapeutics to control tumor growth. Together, we offer an integrative platform to leverage the catalytic activity of endogenous TYR to generate therapeutics through in situ bioorthogonal chemistry for treating melanin-associated skin diseases.

Tyrosinase (TYR) is a crucial and rate-limiting type-3 copper-containing oxidoreductase and mainly participates in two distinct reactions of melanin synthesis[1,2]. TYR presents a dual-core copper (Cu) center structure, which is TYR's active site that binds to histidine residues. According to the difference in the structure of binuclear Cu sites, oxidation state (Eoxy), deoxygenation state (Edeoxy) and reduction state (Emet) forms of catalytic sites have been generated in the process of melanin synthesis[3]. The binuclear Cu sites are vital for the catalytic activity of TYR[4]. TYR is expressed only by melanocytes which synthesize melanin within subcellular organelles called melanosomes, where the Cu-binding domain of TYR is located. These mature melanosomes are transported to the epidermal keratinocytes, resulting in skin pigmentation[5-7]. Melanin overexpression, accumulation, and abnormal reduction in melanocytes could cause the incidence of various diseases, particularly skin diseases, such as skin pigmentation, melanoma, albinism, and vitiligo[8-12]. Given TYR's paramount importance in the melanogenesis pathway, targeting TYR or developing TYR-based therapeutics for regulating melanin production is a promising approach for treating melanin-associated skin diseases.

TYR has up-regulated activity and expression in malignant melanocytes, offering an effective target for developing diagnosis methods

[1]Pharmaceutical Sciences Division, School of Pharmacy, University of Wisconsin-Madison, Madison, WI, USA. [2]Carbone Cancer Center, School of Medicine and Public Health, University of Wisconsin-Madison, Madison, WI, USA. [3]Wisconsin Center for NanoBioSystems, School of Pharmacy, University of Wisconsin-Madison, Madison, WI, USA. [4]Department of Biochemistry, College of Agriculture and Life Sciences, University of Wisconsin-Madison, Madison, WI, USA. ✉e-mail: qhu66@wisc.edu

and treatment strategies against skin diseases. Current strategies in leveraging TYR as a therapeutic target mainly focus on synthesizing and applying small-molecule inhibitors to suppress TYR activity, which results in decreased melanin generation to alleviate skin diseases[13,14]. However, the inherent drawbacks of these inhibitors strongly limit their applications, such as insufficient affinity and selectivity for TYR, potential drug resistance, and dosing-associated toxicity. In this study, to overcome the challenges in developing TYR-based therapeutics for skin disease treatments, we designed a multifaceted strategy to leverage the catalytic activity of TYR to generate therapeutics in situ through the CuAAC reaction. CuAAC, one of the classic click reactions in bioorthogonal chemistry, enables precise in situ catalysis within living systems. Its high selectivity and efficiency facilitate chemical synthesis[15,16] and prodrug activation[17], holding great potential to reduce off-target toxicity and address multidrug resistance[18,19]. Recent developments in CuAAC have shifted towards endogenous Cu catalysts, which catalyze chemical reactions without adding exogenous Cu, mitigating the risk of perturbing copper homeostasis and toxicity against normal tissues[20–22].

To fully unleash the catalytic potential of TYR to develop therapeutics against melanin-associated skin diseases, two treatment strategies were designed (Fig. 1a). First, a TYR degrader was generated by in-situ TYR-catalyzed synthesis of proteolysis-targeting chimeras (PROTACs). We have confirmed that these TYR-catalyzed in-situ formed PRTOACs could effectively degrade overexpressed TYR in the melanocytes and subsequently decrease the melanin levels for alleviating skin hyperpigmentation within 48 h on a mouse model. Second, anti-cancer prodrugs were activated in situ by TYR catalysis to overcome the drug resistance in the current melanoma treatment. We have validated that this TYR-catalyzed prodrug activation approach could effectively retain the intracellular drug concentration to resensitize the drug-resistant melanoma cells to chemotherapeutics both in vitro and in vivo. Collectively, to facilitate the TYR-based therapeutic strategies to treat melanin-associated skin diseases, we here presented a multipronged approach to leverage the catalytic activity of TYR for in situ generation of therapeutics to overcome the challenges in current treatments. The parallel application of these therapeutic methods could offer inspiration from diverse perspectives to design new TYR-based drugs.

## Results

### Catalytic performance of TYR in vitro

To investigate TYR's bioorthogonal catalysis capability in vitro, non-fluorescent precursors 3-azido-7-hydroxycoumarin (pS1) and phenylacetylene (pS2) were selected (Fig. 1b and Supplementary Fig. 1). Upon introduction of mushroom TYR (mTYR), cyan-blue fluorescence ($\lambda$ex = 330 nm, $\lambda$em = 470 nm) from the resulted fluorescent triazole (Fluo) was detected immediately (Supplementary Fig. 2). The fluorescence intensity was significantly enhanced by the addition of sodium ascorbate (SA) (Fig. 1c), outperforming the traditional catalyst, copper sulfate/SA ($CuSO_4$/SA) (Supplementary Fig. 3). Meanwhile, we used high-performance liquid chromatography (HPLC) to quantify the generated Fluo under different treatments, which showed the more Fluo production in TYR-catalyzed group compared to copper sulfate/SA group (Supplementary Fig. 4). Besides this higher catalytic efficiency, using intracellular TYR as the catalyst also has less cytotoxicity against cells than traditional copper sulfate/SA.

Next, we assessed the amount of TYR in normal melanocytes and TYR-overexpressing cell lines. The results suggest that both A375 and B16F10 cells overexpressed TYR, which is about 1.4-fold and 1.8-fold higher than that in normal melanocytes, respectively (Supplementary Fig. 5). To further evaluate the catalytic capability of TYR, a confocal laser scanning microscope was used to observe the generation of fluorescence signals with the treatment of precursors pS1 and pS2. As shown in Supplementary Fig. 6, the barely seen blue

fluorescence was displayed in normal melanocytes with relatively lower TYR expression. Besides, both A375 and B16F10 cells treated with precursors pS1 and pS2 exhibited weak blue fluorescence (Fig. 1d, e). However, adding SA could boost the catalytic efficiency, as evidenced by substantially increased fluorescence. Similar results were further demonstrated by flow cytometry assay, in which higher fluorescence intensities were observed in pS1 + pS2 + SA treatment groups in both A375 and B16F10 cells (Fig. 1f, g). In addition, in-situ TYR-catalyzed synthesis of Fluo was observed under CLSM (Supplementary Fig. 7). This colocalization of Fluo fluorescence with intracellular TYR distribution in A375 and B16F10 cells further verified in-situ click reaction occurred on TYR. Mechanistically, this increase in the catalytic efficacy was attributed to enhanced conversion from *met*-tyrosinase to *deoxy*-tyrosinase by SA[23]. SA reduces the resulting o-quinone back to o-diphenol, effectively regenerating the substrate and facilitating the transition of met-tyrosinase to deoxy-tyrosinase[24,25]. The copper centers in met- and deoxy-tyrosinase exist in Cu (II) and Cu (I) states, respectively[26]. SA enhances catalytic performance in Cu (I)-driven click reactions by stabilizing the active Cu (I) state. Collectively, these results validated the robust catalytic potential of endogenous TYR.

### Mechanistic studies of TYR's catalytic activity in CuAAC reaction

To further decipher the catalytic performance of TYR, the ab initio molecular dynamics (AIMD) was applied to investigate the geometric and electronic properties of TYR and its catalytic effect on the reaction between the azido group and ethynyl group[27,28]. First, the simulation model containing the activating center (~400 atoms) of TYR was constructed (Supplementary Fig. 8). According to Einstein's relations, 10 ps was selected as an equilibrium of the truncated model of the activation center of TYR to obtain the simulated diffusion coefficient (D) at 298 K by using AIMD simulation. Next, the azido and ethynyl molecules were simplified to reduce complexity by relaxing to a local minimum configuration of the models for AIMD and Metadynamics simulation, meanwhile keeping the functional group of the reaction. The radial distribution function of atoms among Cu, N, and C atoms was shown in Fig. 2a; the Cu-Cu distance distribution had two main peaks (about 4 and 5 Å), implying a flexible distance. This flexibility of Cu-Cu indicates its available space for the reactant to bind and activate. However, the main peak at about 2 Å and 3 Å of the Cu-N and Cu-C distribution implied the rigid framework of activation sites of Cu which can prevent their detachment. The exchange-correlation energy was calculated using the generalized-gradient approximation, specifically the Perdew, Burke, and Ernzerhof (PBE) parameterization[29,30]. The charge density differences of azido and ethynyl were vertically bound to two Cu activation sites, representing the coordinative bonding interaction (Fig. 2b, c), which was in accordance with the interaction of Cu between N and C atoms in Cu-catalyzed click reaction. Subsequently, the process of the TYR-catalyzed azido and ethynyl molecules from an intermediate to the final product was shown in Fig. 2d by the geometries of the Metadynamics simulation model. In the Metadynamics simulation, the distance between the two N atoms of the azido group and the C atoms of ethynyl was used as the collective variables to reduce the dimension of the reaction space to calculate the free energy. We performed well-tempered single-walker metadynamics simulations and carried out all simulations at an elevated temperature of 298 K[31,32]. The result of the energy variation from TYR-catalyzed azido and ethynyl molecules to the intermediate (Fig. 2e) revealed that this step was an exothermic reaction, which released ~2.73 eV and had a barrier of 0.43 eV that made this reaction occur very easily. Notably, the energy change from the intermediate response to the product described a low barrier of 0.32 eV (Fig. 2f), validating that this is a more facile reaction than the previous one. Collectively, TYR has robust catalytic activity in the CuAAC reaction.

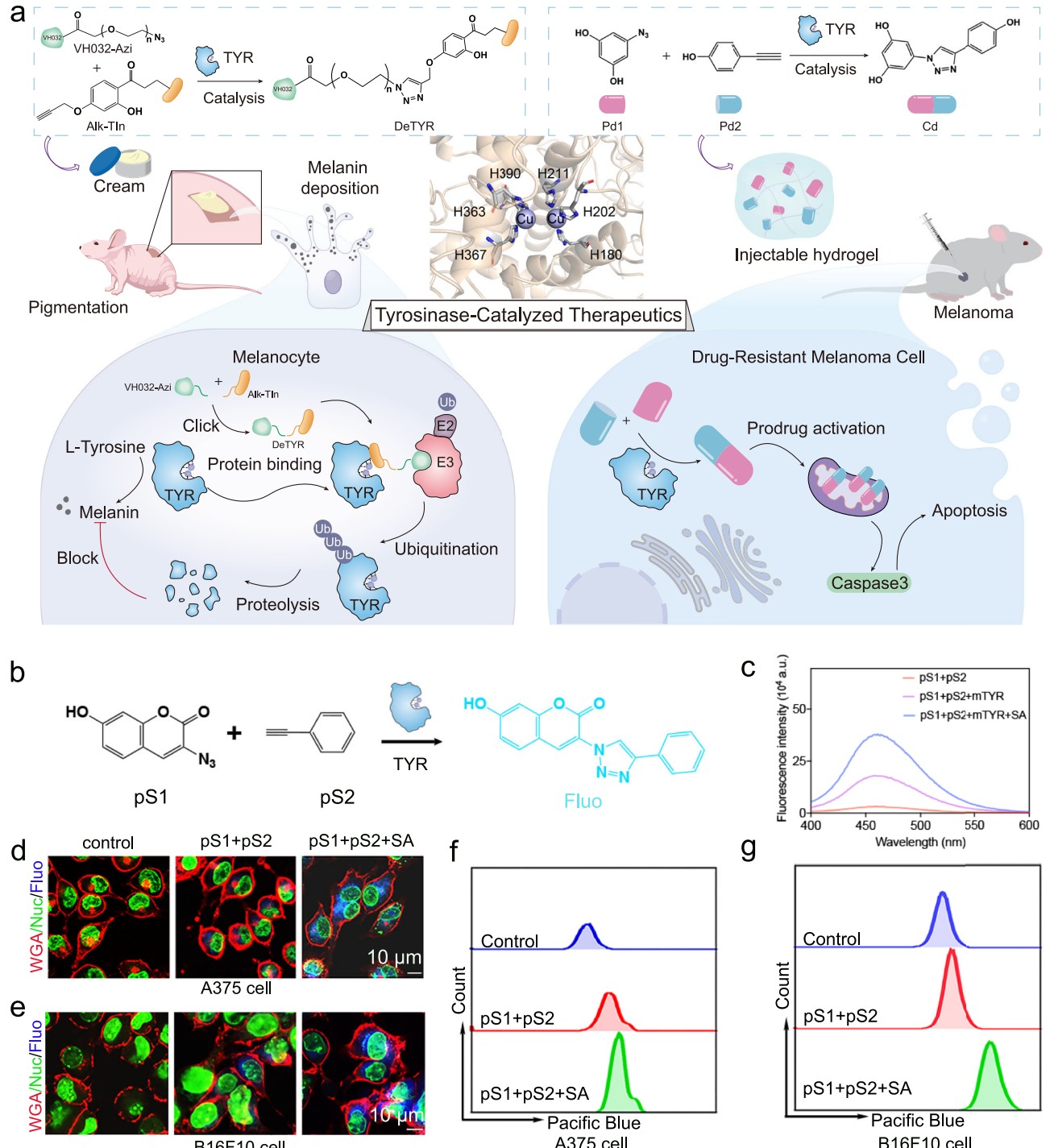

**Fig. 1 | TYR-catalyzed therapeutics. a** Schematic of the design of TYR-catalyzed therapeutics. Left: TYR-catalyzed in-situ clicked PROTACs to degrade TYR for treating skin hyperpigmentation. Right: TYR-catalyzed in-situ prodrug activation for treating drug-resistant melanoma. **b** Scheme of TYR-catalyzed fluorescent precursors (pS1 and pS2) via the CuAAC reaction. **c** Fluorescence spectra of pS1 + pS2, pS1 + pS2 + mTYR, and pS1 + pS2 + mTYR + SA at the concentration of 10 μM. a. u., arbitrary units. (mTYR:0.1 mg/mL). Representative confocal microscopy images of A375 (**d**) and B16F10 (**e**) cells treated with pS1 + pS2 and pS1 + pS2 + SA, respectively. (*n* = 3 independent experiments). Flow cytometry assays of A375 (**f**) and B16F10 (**g**) cells treated with pS1 + pS2 and pS1 + pS2 + SA, respectively.

## TYR-catalyzed in-situ formed TYR degraders

Given the critical role of TYR in melanin production and melanin-related skin diseases, we seek to develop therapeutics centralizing TYR as the target in this study. The most commonly used approach to target TYR is to apply selective inhibitors to suppress the overactivity or overexpression of TYR through binding to the free enzyme or enzyme-substrate complex. However, traditional small-molecule-based inhibitors exhibit the functionality by a robust and tight occupancy of the enzyme for a long time, which could induce the concern of overdosing toxicity due to their lack of recyclability. Recently, an emerging approach to inhibit overactive or over-expressed proteins is to develop proteolysis-targeting chimeras (PROTACs) that can efficiently degrade the targeted protein of interest (POI) by hijacking the ubiquitin-proteasome system[33,34]. PROTACs, heterobifunctional molecules comprised of functional ligands for POI and E3 ligases that are bridged by a linker, offer robust protein

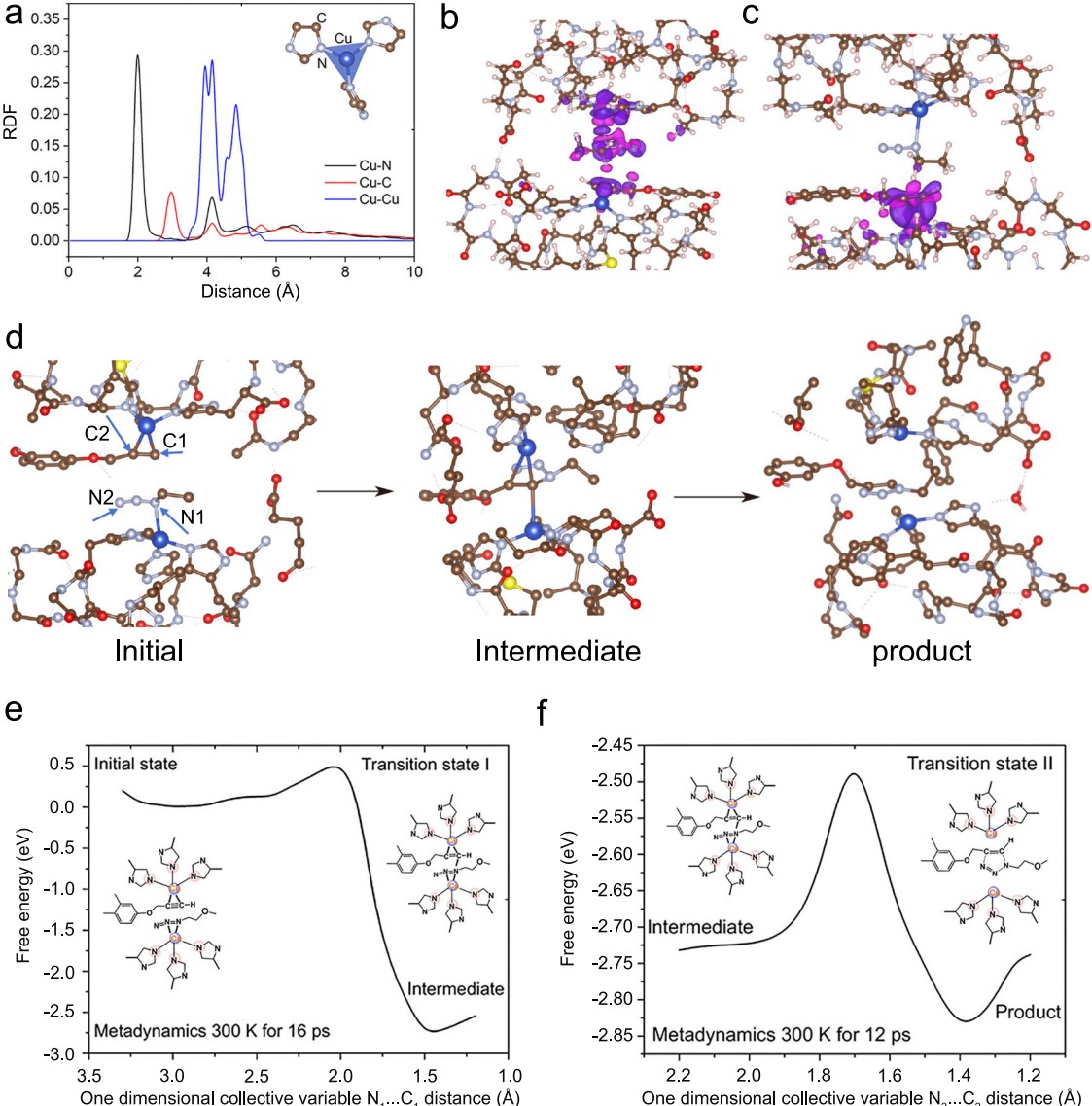

**Fig. 2 | AIMD studies of the TYR-catalyzed azide-alkyne cycloaddition reaction.**
**a** Radial distribution function of atoms between Cu-N, Cu-C and Cu-Cu atoms. Atoms are color-coded: carbon in brown, copper in blue, nitrogen in white, and oxygen in red. Charge density difference of azido (**b**) and ethynyl (**c**) binding vertically on two Cu activation sites representing the coordinative bonding interaction, respectively. **d** Scheme of catalytic transformation from the initial state to the intermediate to the final product. **e** Energy variation of TYR-catalyzed two

individual azido and ethynyl molecules to an intermediate. Simulation condition: one-dimensional free energy surface from a metadynamics simulation at 298 K for 16 ps. **f** Energy change of reaction from intermediate to final product. Simulation condition: one-dimensional free energy surface from a metadynamics simulation at 298 K for 12 ps. Insert in (**e**, **f**) is the Lewis structure of azido and ethynyl binding to TYR at different transition states.

degradation/inhibition efficacy surpassing small-molecule inhibitors[35], primarily owing to their catalytic nature[36]. Unlike occupancy-driven pharmacology, PROTACs function actively regardless of target occupancy and can be recycled to degrade additional POIs after each cycle of ubiquitination and POI degradation[37,38]. Their mechanism creating a transient ternary complex with POIs, E3 ligases, and PROTACs minimizes susceptibility to protein mutations and enhances their potential to overcome drug resistance[39]. Intrigued by the advantages of PRO-TACs in degrading dysfunctional proteins, herein, we devised an in-situ formed TYR degrader by utilizing the potent catalytic performance of TYR (Fig. 3a). First, we introduced an alkyne moiety to the TYR inhibitor (TIn) and compared the inhibition kinetics of Alk-TIn and TIn against TYR[40]. Both TIn and Alk-TIn strongly inhibited the activity of mTYR with $IC_{50}$ values of 51.1 nM and 92.6 nM, respectively, which were both much lower than the commercial TYR inhibitor kojic acid (34.1 μM) (Supplementary Fig. 9). The kinetic data, which is presented

as Lineweaver-Burk double reciprocal plots, revealed four linear relationships between 1/V and 1/[S], with varying slopes that intersect along the horizontal axis (Supplementary Fig. 10). The maximum velocity (Vmax) decreased along with the increase of Alk-TIn concentrations, while the Michaelis constant (Km) did not change. Taken together, Alk-TIn functions as a noncompetitive inhibitor of mTYR, possibly by engaging amino acids outside the active binding site. Subsequently, we designed the VHL E3 ligase-recruiting ligand VHO32 for the in-situ construction of TYR-degrading PROTACs. Three azido-modified VHO32 ligands were synthesized by adjusting the lengths of linkers, named VHO32-Azi1, VHO32-Azi2, and VHO32-Azi3, respectively. We hypothesized that the Alk-TIn would bind to the intracellular TYR and be further clicked with modified VHO32 under the catalysis of TYR to construct an in-situ formed PROTACs to degrade TYR.

Before validating the TYR degradation, we first evaluated the cytotoxicity of synthesized compounds against human A375 and

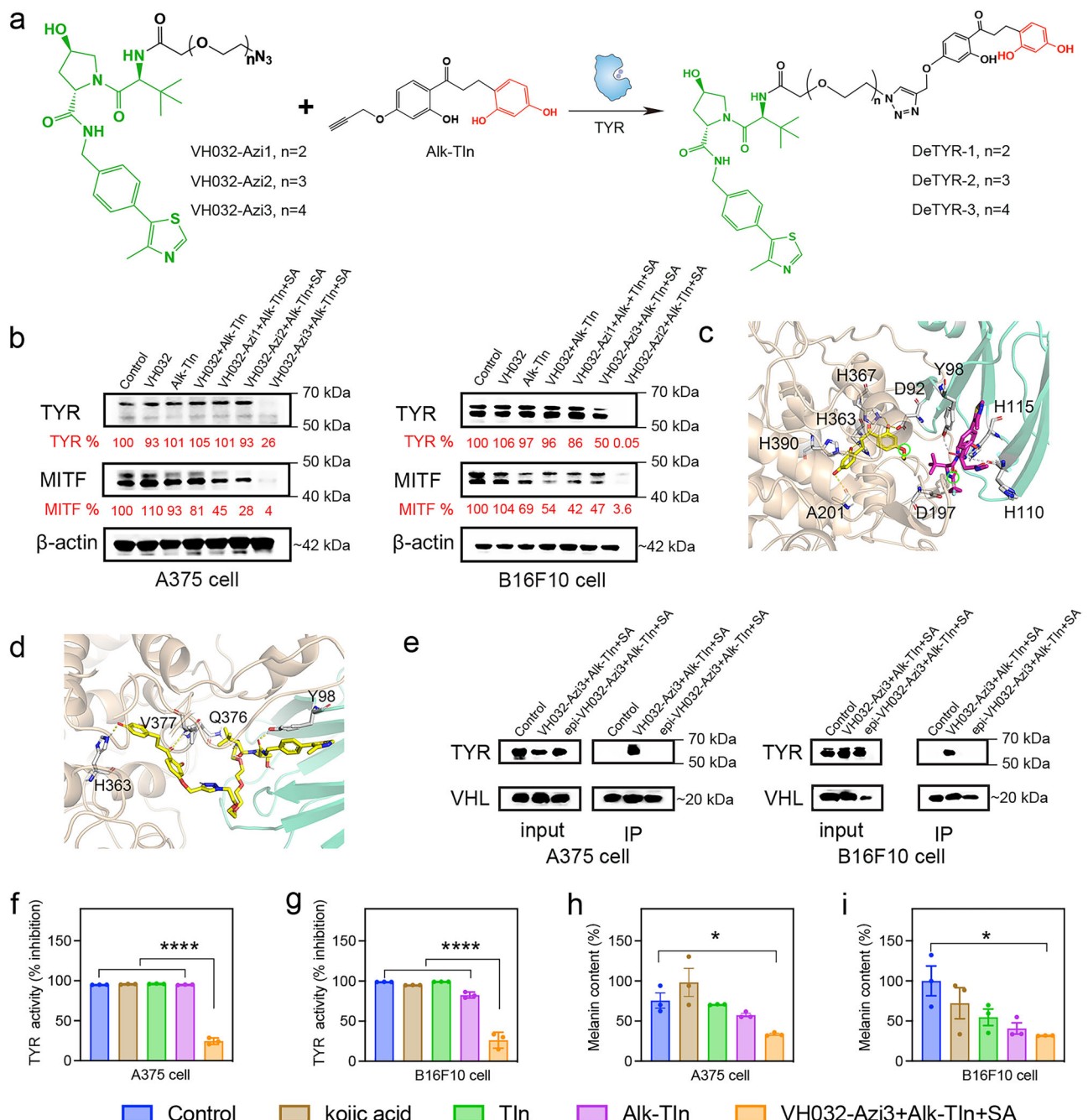

**Fig. 3 | TYR-catalyzed in-situ formed TYR degraders. a** Chemical structure of azido-modified VH032 ligands (VH032-Azi1, VH032-Azi2, VH032-Azi3); alkyne-modified TYR inhibitor (Alk-TIn), and click-catalyzed TYR degraders (DeTYR-1, DeTYR-2, DeTYR-3). **b** Western blot analysis of TYR and MITF in A375 and B16F10 cells treated with VH032, Alk-TIn, VH032 + Alk-TIn, VH032-Azi1 + Alk-TIn + SA (0.5 μM), VH032-Azi2 + Alk-TIn + SA (0.5 μM) and VH032-Azi3 + Alk-TIn + SA (0.5 μM) at the concentration of 0.1 μM for 24 h. ($n$ = 3 independent experiments). **c** Molecular dynamic simulation of TYR (wheat), Alk-TIn (yellow), VHL (cyan), and VH032-Azi3 (magenta). The distance was measured by the two linker atoms (circled in green). **d** Molecular dynamic simulation of DeTYR-3 (yellow) interacting with VHL (cyan) and TYR (wheat) proteins. **e** Co-immunoprecipitation of TYR and VHL

from lysates of A375 and B16F10 cells treated with VH032-Azi3 (0.1 μM) + Alk-TIn (0.1 μM) + SA (0.5 μM) or epi-VH032-Azi3 (0.1 μM) + Alk-TIn (0.1 μM) + SA (0.5 μM) for 4 h. ($n$ = 3 independent experiments). TYR inhibition of blank control, kojic acid, TIn, Alk-TIn and VH032-Azi3 + Alk-TIn + SA (0.5 μM) at 0.1 μM in A375 (**f**) and B16F10 (**g**) cells, respectively. ($n$ = 3 biological replicates). (****$P$ < 0.0001). Data are presented as mean ± standard deviation (SD). Melanin content of blank control, kojic acid, TIn, Alk-TIn, and VH032-Azi3 + Alk-TIn + SA at 0.1 μM in A375 (**h**) ($P$ = 0.0479) and B16F10 (**i**) ($P$ = 0.0147) cells, respectively. ($n$ = 3 biological replicates). Data are presented as mean ± standard error (SEM). Statistical analysis was performed using ONE-WAY variance (ANOVA) by Dunnett's multiple comparisons (*$P$ < 0.05).

murine B16F10 melanoma cells, which are both TYR-overexpressing cell lines. As shown in Supplementary Figs. 11, 12, all tested compounds, including VH032 + Alk-TIn, VH032-Azi1 + Alk-TIn, VH032-Azi2 + Alk-TIn, and VH032-Azi3 + Alk-TIn, displayed negligible cytotoxicity against both melanoma cell lines under the 25 μM concentration. Next,

to screen the best PROTACs for TYR degradation, we performed western blot (WB) analysis to evaluate the degradation efficiency of TYR and the associated expression level of microphthalmia-associated transcription factor (MITF), a key transcriptional activator in melanin synthesis. Both A375 and B16F10 cells were treated with VH032, Alk-

TIn, VH032 + Alk-TIn, VH032-Azi1 + Alk-TIn + SA, VH032-Azi2 + Alk-TIn + SA, and VH032-Azi3 + Alk-TIn + SA at 0.1 μM for 12 and 24 h. The VH032-Azi3 + Alk-TIn + SA treatment group displayed the highest degradation efficiency of TYR among all treatment groups in both A375 and B16F10 cells, in which 24 h (Fig. 3b) treatment showed better efficacy than 12 h (Supplementary Fig. 13). Notably, in B16F10 cells, 24 h treatment with VH032-Azi3 + Alk-TIn + SA group could degrade nearly 100% of TYR. In addition, the best TYR degradation efficacy was validated by TYR immunocytochemical staining, in which the lowest fluorescence signal was found in VH032-Azi3 + Alk-TIn + SA treatment group in both A375 and B16F10 cells (Supplementary Figs. 14, 15). To further explore the TYR degradation performance of VH032-Azi3 + Alk-TIn + SA treatment group, we performed the dose- and time-dependent experiments, in which higher doses and longer treatment time resulted in better degradation efficiency (Supplementary Figs. 16, 17). As shown in Supplementary Fig. 18, the WB analysis suggested the in-situ formed PROTACs could continuously degrade TYR at least for 28 h and 32 h in A375 and B16F10 cells, respectively. Collectively, the DeTYR-3 resulted from the TYR-catalyzed VH032-Azi3 + Alk-TIn + SA treatment group displayed the best TYR degradation efficacy and was selected for the following mechanistic exploration and in vivo studies.

## Mechanism of Action (MoA) exploration of TYR-catalyzed in-situ formed TYR degraders

To further delve into the underlying MoA of TYR degraders, we first detected the intracellular presence of DeTYR-3. As displayed in Supplementary Figs. 19, 20, the existence of DeTYR-3 was validated in both A375 and B16F10 cells by LC-MS analysis. In addition, we applied silico modeling molecular docking to evaluate the protein binding of VH032-Azi3, Alk-TIn, and DeTYR-3 to TYR and VHL E3 ligases, guided by the crystal structures of TYR and VHL. As shown in Fig. 1a, the structure of TYR displays six histidine amino acid residues around the active position of two copper ions: His367, His363, His390, His211, His202, and His180. First, the Alk-TIn complex was constructed by employing Autodock vina to carry out molecule docking[41] (Supplementary Fig. 21). Next, the complex model for VH032-Azi3 and Alk-TIn was established using global protein-protein docking with PatchDock[41,42] (Supplementary Fig. 22). Based on Rosetta protein−protein docking protocol, the decoy with the distance between VH032-Azi3 and Alk-TIn atoms was less than 21 Å (Fig. 3c), occupying significant cavities within the VHL E3 ligase and TYR. Hydrogen bonding was observed between Alk-TIn and residues H367, H363, H390, A201 and between VH032-Azi3 and residues Y98, H115, H110, and D197. Importantly, neither the azido nor ethynyl groups interacted with residues, affirming the feasibility of the click reaction. Subsequently, the qualified interaction modes with the minimum ligand conformer energy and protein docking score were used to construct optimal VHL-DeTYR-3-TYR ternary complex[43] through pairwise RMSD analysis[44] (Supplementary Fig. 23). The docking profile identified the most stable VHL-DeTYR-3-TYR ternary complex (Fig. 3d). Specifically, the Alk-TIn moiety bound to H363 and V377 of TYR, and the VH032-Azi3 moiety bound to Y98 and Q376 of VHL, both of which were through hydrogen bonds. These interactions and conformations closely mirrored those observed in the original crystal structures, indicating that VH032-Azi3 and Alk-TIn could form a stable ternary complex with VHL and TYR.

Next, we applied a co-immunoprecipitation assay to validate the formation of the stable ternary complex. An inactive control of VH032 was synthesized (designated epi-VH032), which has an identical physicochemical structure to VH032 but cannot bind to VHL E3 ligases. As shown in Fig. 3e, in both A375 and B16F10 cells, only DeTYR-3 formed from the TYR-catalyzed VH032-Azi3 + Alk-TIn + SA group could successfully form a ternary complex between TYR, DeTYR-3, and VHL, as evidenced by the detected TYR in the final precipitated protein samples pulled down by anti-VHL antibodies in VH032-Azi3 + Alk-TIn + SA group rather than DMSO and epi-VH032-Azi3 + Alk-TIn + SA control

groups. To further substantiate the mechanism of proteasome-mediated TYR degradation, the proteasome inhibitor epoxomicin (Epox) was applied. The addition of Epox fully abolished the TYR degradation in the VH032-Azi3 + Alk-TIn + SA treatment group, suggesting that TYR degradation is through the UPS mechanism (Supplementary Fig. 24).

To evaluate the impact of TYR degradation on TYR activity, TYR catalyzation against the substrate L-dopa was evaluated after various treatments. The VH032-Azi3 + Alk-TIn + SA treatment group significantly decreased TYR activity in both A375 and B16F10 cells (Fig. 3f, g), primarily due to the substantial degradation of TYR. Furthermore, the melanin contents were reduced due to TYR degradation in the VH032-Azi3 + Alk-TIn + SA treatment group (Fig. 3h, i).

## Therapeutic efficacy evaluation of TYR degradation in a mouse hyperpigmentation model

To demonstrate the in vivo therapeutic effect of in-situ TYR-catalyzed PROTAC formation against skin disease, the skin penetration capability of the cream containing VH032-Azi3 and Alk-TIn was first explored. As shown in Supplementary Fig. 25, deeper penetration of the FAM-VH032-Azi3 and Cy5.5-Alk-TIn in the cream group was supported by the fact that the fluorescence group would appear in the epidermis and upper dermis, while it only appeared in the epidermis in the control group.

After validation of effective degradation of TYR and subsequent melanin decrease, we evaluated the in vivo treatment efficacy on a mouse skin hyperpigmentation model, which was established by the implementation of 1,3-dihydroxyacetone (DHA)-based self-tanning drops and UV irradiation. Next, we developed a cream formulation encapsulating seven different therapeutics (Supplementary Fig. 26), including (1) blank control, (2) VH032, (3) Alk-TIn, (4) VH032 + Alk-TIn, (5) epi-VH032-Azi3 + Alk-TIn + SA, (6) DeTYR-3 and (7) VH032-Azi3 + Alk-TIn + SA. The creams with good moisturization and biocompatibility were used to treat the mice for two days consecutively (Fig. 4a). The color changes of square hyperpigmentation patterns on the back of the mice were observed on a daily basis. As shown in Fig. 4b and Supplementary Fig. 27, both blank cream and VH032 cream did not show any therapeutic efficacy against hyperpigmentation during the treatment course. A moderate fading in the pigmentation patterns was observed in Alk-TIn cream and VH032 + Alk-TIn cream groups, which was attributed to the TYR inhibition capability of Alk-TIn. The epi-VH032-Azi3 + Alk-TIn + SA treatment group demonstrated negligible efficacy due to the incapability of VHL binding of epi-VH032-Azi3, which is pivotal for recruiting E3 ligase and degrading TYR. The DeTYR-3 group displayed an improved treatment outcome in attenuating the hyperpigmentation. However, all mice treated with VH032-Azi3 + Alk-TIn + SA creams showed a notable and rapid disappearance of pigmentation within 48 h, indicating the substantial improvement of these hyperpigmentation disorders. In addition, the skin tissues were collected for the Fontana-Massion staining to further analyze the hyperpigmentation levels. As displayed in Fig. 4c, d, VH032-Azi3 + Alk-TIn + SA cream treatment resulted in the lowest pigment deposition among all treatment groups, which was a 2.5-fold decrease than the blank cream treatment group, a 2.1-fold decrease than the epi-VH032-Azi3 + Alk-TIn + SA cream treatment group, and a 1.3-fold decrease than the DeTYR-3 group. To further investigate the underlying mechanism for this improved hyperpigmentation therapeutic efficacy, we first assessed the melanin levels in mouse skin tissues following various treatments. As shown in Fig. 4e, melanin levels remained largely unchanged in the blank and VH032 groups, and modest decreases were found in the Alk-TIn, VH032 + Alk-TIn, and epi-VH032-Azi3 + Alk-TIn + SA treatment groups. A pronounced reduction in melanin levels was observed in the VH032-Azi3 + Alk-TIn + SA cream group as a consequence of TYR degradation, which showed a 2.8-fold decrease than the blank cream treatment group, a 2.2-fold decrease than the epi-

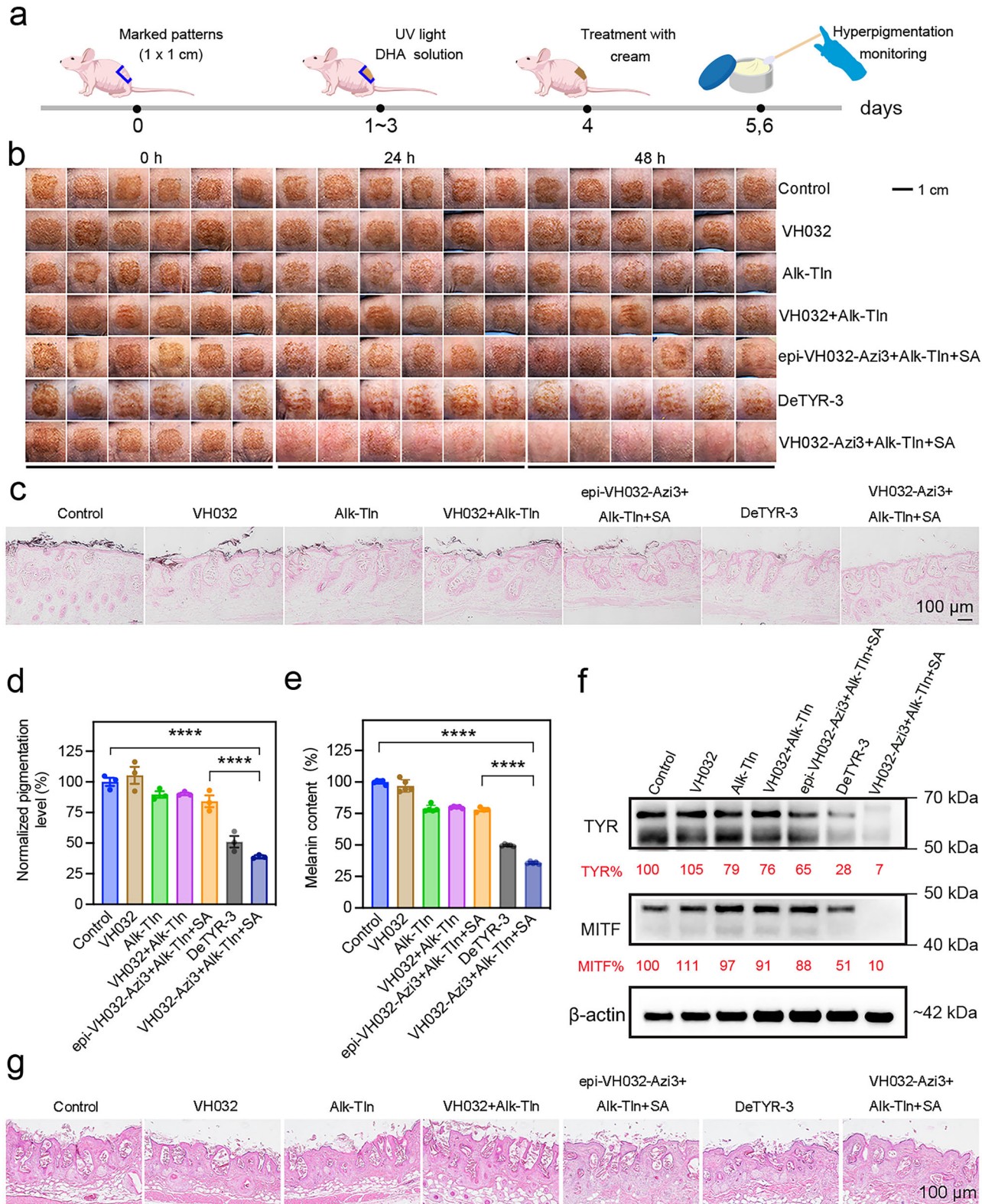

**Fig. 4 | Therapeutic efficacy evaluation of TYR degradation in a mouse hyperpigmentation model. a** Schematic of the experimental plan in the mouse hyperpigmentation model. **b** Images of square pigmentation patterns on the backs of mice treated with (1) blank control, (2) VH032, (3) Alk-Tln, (4) VH032 + Alk-Tln, (5) epi-VH032-Azi3 + Alk-Tln + SA, (6) DeTYR-3 and (7) VH032-Azi3 + Alk-Tln + SA creams at different time points (0, 24 and 48 h). **c** Fontana-Masson (FM) staining of melanin in mouse skin. (*n* = 3 independent samples). **d** Normalized pigmentation level in different treatment groups by digital image analysis using Image J. (*n* = 3

biological replicates). Data are presented as mean ± standard deviation (SD). ****P < 0.0001. **e** Melanin contents of mouse skin samples treated with different groups for 48 h. (*n* = 5 biological replicates). Data are presented as mean ± standard deviation (SD). ****P < 0.0001. **f** Western blot assay of TYR and MITF in mice skin samples treated with different groups for 48 h. (*n* = 3 independent experiments). **g** H&E staining of mouse skin tissues treated with different groups for 48 h. (*n* = 3 independent samples). Statistical analysis was performed using ONE-WAY variance (ANOVA) by Dunnett's multiple comparisons.

VH032-Azi3 + Alk-TIn + SA cream treatment group, and a 1.3-fold decrease than the DeTYR-3 group. In addition, the TYR expression level in the treated skin tissues was evaluated by the western blot assay. Negligible and moderate changes in TYR expression were observed in blank and VH032 cream treatment groups, and Alk-TIn, VH032 + Alk-TIn, and epi-VH032-Azi3 + Alk-TIn + SA treatment groups, respectively. In contrast, a substantial decrease in TYR levels was found in the VH032-Azi3 + Alk-TIn + SA cream treatment group (Fig. 4f). Finally, the potential side effects of all treatment groups were examined by hematoxylin and eosin (H&E) staining of treated skins, which all showed negligible damage against skin tissues as evidenced by preserved epidermal integrity and no sign of immune cell infiltration (Fig. 4g). Notably, as displayed in Supplementary Fig. 28, we did not see any recurrence of hyperpigmentation over the observation course of 30 days in the VH032-Azi3 + Alk-TIn + SA cream treatment group, substantiating long-term efficacy. Collectively, these TYR-catalyzed in-situ formed PROTACs could effectively degrade overexpressed TYR in the melanocytes and subsequently decrease the melanin levels for quick alleviation of hyperpigmentation.

### TYR-catalyzed in-situ activation of anti-cancer prodrugs
Besides hyperpigmentation caused by overexpressed or overactivated TYR, another common disease associated with dysfunctional TYR is melanoma, which is characterized by excessive melanin synthesis catalyzed using TYR. Chemotherapy, one of the major treatment options for cancer, has limited efficacy in melanoma therapy largely due to the drug resistance followed by melanoma relapse after the initial response. Bioorthogonally activated prodrugs can overcome drug resistance by synthesizing anti-cancer drugs intracellularly. Herein, to explore the potential of TYR as a catalyst to activate anti-cancer prodrugs against melanoma drug resistance, we designed a clickable prodrug approach in which drugs will be only toxic against melanoma cells after TYR catalysis.

Prodrug 1 (Pd1; 5-azidobenzene-1,3-diol) and prodrug 2 (Pd2; 4-ethynylphenol) were synthesized first. TYR catalyzed two prodrugs by a click reaction to form a cancer drug (designated Cd: 5-(4-(4-hydroxyphenyl)-1H-1,2,3-triazol-1-yl) benzene-1,3-diol), which is a resveratrol derivative[45] (Fig. 5a). To study intracellular TYR-catalyzed prodrug activation and its efficacy against drug resistance, drug-resistant A375 and B16F10 cell lines (designated DR-A375 and DR-B16F10) were established (Supplementary Fig. 29) according to the reported method[46]. Next, the intracellular construction of Cd was evaluated by LC-MS. As shown in Supplementary Figs. 30, 31, the intracellular presence of Cd due to TYR-catalyzed prodrug activation was detected in both DR-A375 and DR-B16F10 cells. The cytotoxic effects of Pd1 and Pd2 were assessed by a cell counting kit-8 (CCK-8) assay, showing minimal cytotoxicity at a concentration of 100 μM (Supplementary Fig. 32). In contrast, Pd1 + Pd2 + SA treatment resulted in dose-dependent cell toxicity in both DR-A375 and DR-B16F10 cells (Fig. 5b, c). Notably, Cd treatment showed very limited toxicity against DR-A375 and DR-B16F10 cells, validating the drug resistance capability of both cells. In addition, the cancer cell apoptosis was evaluated by flow cytometry after staining with fluorescein Annexin V and propidium iodide. The Pd1, Pd2, and Cd treatments did not induce considerable apoptosis in DR-A375 and DR-B16F10 cells (Fig. 5d and Supplementary Fig. 33). However, there is significantly increased apoptosis in both drug resistance cell lines after Pd1 + Pd2 + SA treatment, which displayed a 5.5-fold increase in DR-A375 and a 4.8-fold increase in DR-B16F10 cells compared to Cd, respectively (Supplementary Fig. 34).

To explore the potential underlying mechanism for overcoming drug resistance of the TYR-catalyzed prodrug approach, we evaluated the intracellular drug concentration in Cd and Pd1 + Pd2 + SA treatment groups. Drug efflux pumps, a key mechanism in drug resistance, reduce intracellular drug concentrations, diminishing the effectiveness of chemotherapeutics[47]. We used HPLC to measure intracellular

and extracellular Cd concentration after the treatment with Cd alone or with Pd1 + Pd2 + SA in DR-A375 and DR-B16F10 cells at 24 and 48 h. Notably, in DR-A375 cells treated with Pd1 + Pd2 + SA, no extracellular Cd was detected at 24 h, and minimal efflux was observed at 48 h, with over 75% of Cd retained intracellularly (Fig. 5e). In contrast, the Cd treatment group exhibited significant drug efflux, with about 50% and 20% of Cd retained intracellularly at 24 and 48 h, respectively. In DR-B16F10 cells, no extracellular Cd was detected in the Pd1 + Pd2 + SA treatment group at 24 and 48 h (Fig. 5f). However, the Cd treatment group retained ~20% of intracellular Cd. Collectively, these results indicated that drug-resistance cells could leverage efflux pumps to expel endocytosed drugs, resulting in low intracellular drug concentration to alleviate the cytotoxicity of anti-cancer drugs. Encouragingly, our TYR-catalyzed prodrug approach could evade drug resistance by maintaining intracellular drug concentration through in-situ bioorthogonal chemistry. In addition, since the prodrug is non-toxic before activation, there will be minimal dosing-related toxicity for the prodrug approach, in which a large dose could be used without severe safety concerns to overcome drug resistance.

Finally, a three-dimensional (3D) tumor spheroid model was established to better recapitulate the physiological features of solid tumors. These tumor spheres were exposed to varying concentrations of Cd and Pd1 + Pd2 + SA for 48 h (Fig. 5g, i). Utilizing fluorescence microscopy for volumetric analysis, we observed a substantial increase in spheroid diameter following Cd treatment even at 100 μM for both tested cell lines. In contrast, treatment with Pd1 + Pd2 + SA resulted in a concentration-dependent decrease in spheroid size. Quantitative analysis of growth kinetics for DR-A375 (Fig. 5h) and DR-B16F10 (Fig. 5j) tumor spheroids revealed a marked reduction in spheroid volume under Pd1 + Pd2 + SA treatment, particularly at 100 μM concentrations where the tumor spheroid completely fell apart due to induced cancer cell apoptosis.

### Therapeutic efficacy evaluation of TYR-activated anti-cancer prodrugs in a mouse drug-resistant melanoma model
Encouraged by the strong anti-proliferation effects observed in vitro, we further explored the therapeutic potential of TYR-activated anti-cancer prodrugs on a DR-B16F10 melanoma-bearing mouse model. To facilitate the in vivo administration of various small-molecule therapeutics, we developed a Pluronic™ F-127-based injectable, thermo-responsive hydrogel for peritumoral administration[48]. The hydrogel underwent sol-to-gel transition within one minute at the body temperature (Supplementary Fig. 35). After the size of subcutaneous DR-B16F10 melanoma reached about 100 mm³, the tumor-bearing mice were randomized and treated with various therapeutics every other day three times in total (Fig. 6a). As shown in Fig. 6b, the TYR-activated prodrug group (Pd1-Pd2-SA@Gel) displayed a superior tumor inhibition efficacy compared to other treatment groups. In contrast, the Cd@Gel group failed to control tumor growth due to the drug-resistant nature of DR-B16F10 melanoma. Interestingly, the free drug group (Pd1-Pd2-SA) showed moderate efficacy against DR-B16F10 melanoma, which is mainly attributed to the quick clearance of small molecules after injection, highlighting the advantage of the hydrogel as a depot to control the sustained release of encapsulated small-molecule drugs. In addition, the survival of DR-B16F10 melanoma-bearing mice was monitored during the treatment course. All the mice treated with the therapeutic groups, except for Pd1-Pd2-SA@Gel, died within 21 days due to the aggressive and drug-resistant characteristics of DR-B16F10 melanoma. In sharp contrast, the Pd1-Pd2-SA@Gel group significantly extended the survival of DR-B16F10 melanoma-bearing mice with a median survival time of 27 days (Fig. 6c). Notably, there were no significant fluctuations in body weight for all treatment groups during the treatment course (Fig. 6d). To further confirm the therapeutic efficacy of Pd1-Pd2-SA@Gel, the tumor tissues were collected for TUNEL and H&E staining. Significant apoptosis in tumor cells from

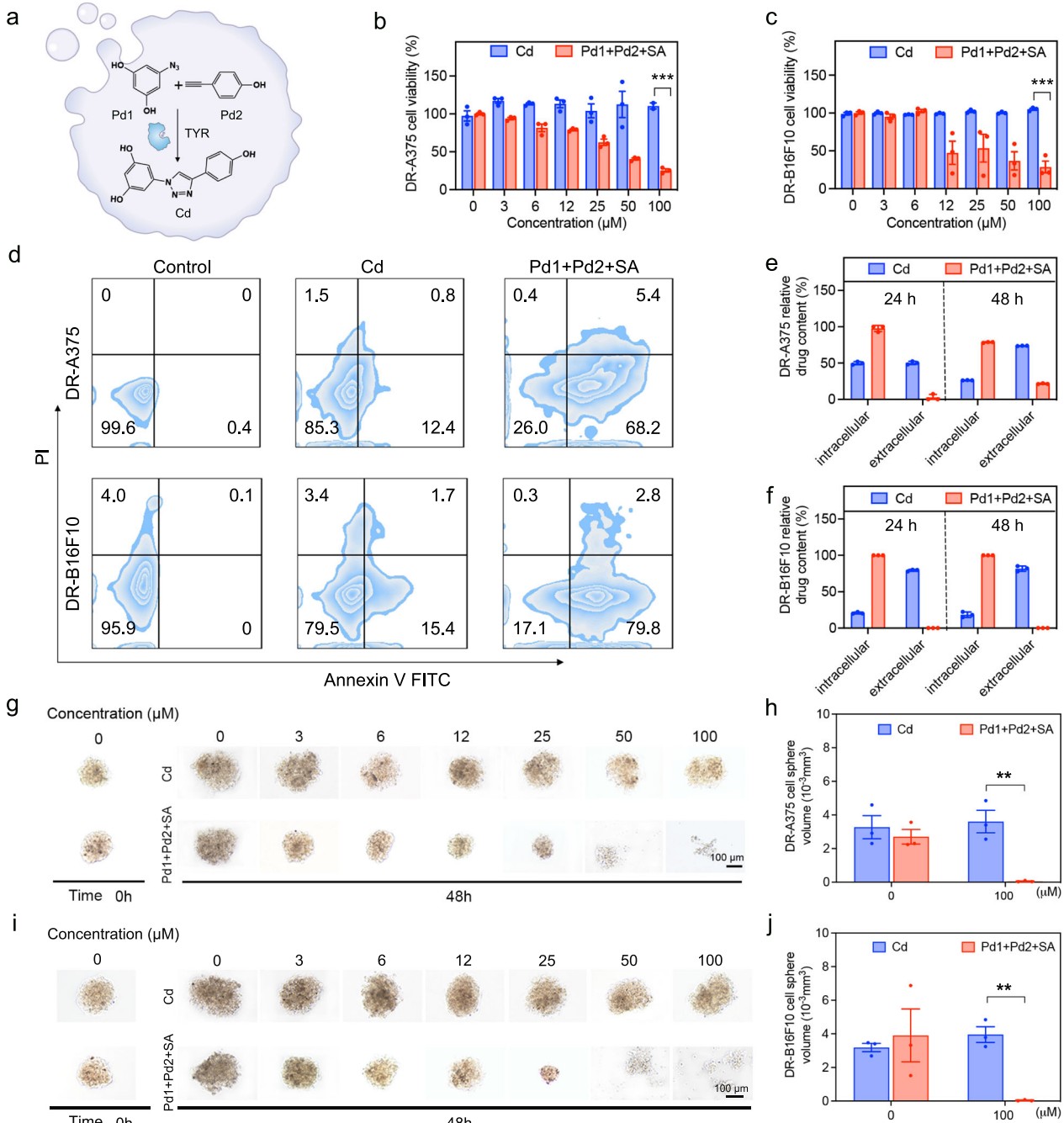

**Fig. 5 | TYR-catalyzed in-situ activation of anti-cancer prodrugs. a** Schematic illustration of TYR-catalyzed intracellular prodrug activation. Cytotoxicity analysis of Cd and Pd1 + Pd2 + SA treatment against DR-A375 (**b**) (*P* = 0.00024) and DR-B16F10 (**c**) (*P* = 0.00063) cells for 24 h, respectively. (*n* = 3 biological replicates). Data are presented as mean ± standard error (SEM). **d** Flow cytometry analysis of the apoptosis of DR-A375 and DR-B16F10 cells after treatments with Cd, and Pd1 + Pd2 + SA for 24 h. High-performance liquid chromatography (HPLC) measurements of intracellular and extracellular Cd after 24 and 48 h incubation with Cd and Pd1 + Pd2 + SA in DR-A375 (**e**) and DR-B16F10 (**f**) cells. (*n* = 3 biological replicates). Data are presented as mean ± standard deviation (SD). **g** Images of DR-A375 tumor spheres treated with Cd and Pd1 + Pd2 + SA under different concentrations for 48 h. **h** Average volume of DR-A375 tumor spheres at 48 h after different treatments. (*n* = 3 biological replicates). Data are presented as mean ± standard error (SEM). *P* = 0.0061. **i** Images of DR-B16F10 tumor spheres treated with Cd and Pd1 + Pd2 + SA under different concentrations for 48 h. **j** Average volume of DR-B16F10 tumor spheres at 48 h after different treatments. (*n* = 3 biological replicates). Data are presented as mean ± standard error (SEM). *P* = 0.0011. Statistical analysis was performed *via* unpaired Student's *t*-test (two-tailed) (**\**P* < 0.01, \***\**P* < 0.001).

the Pd1-Pd2-SA@Gel group was demonstrated by TUNEL staining, suggesting effective induction of apoptotic cell death through TYR-activated prodrugs (Fig. 6e and Supplementary Fig. 36). In addition, H&E staining also validated the enhanced tumor cell death in the Pd1-Pd2-SA@Gel group compared to other treatment groups (Fig. 6f). Finally, the comprehensive safety profile of Pd1-Pd2-SA@Gel treatment was evaluated. As shown in Fig. 6g, the complete blood cell counts revealed negligible changes in various blood cells. Furthermore, the functions of the liver and kidney were not impacted, as evidenced by insignificant changes in liver function marker ALT and AST and kidney function marker CREA after the treatment of Pd1-Pd2-SA@Gel (Fig. 6h). Besides, the major organs were collected for H&E

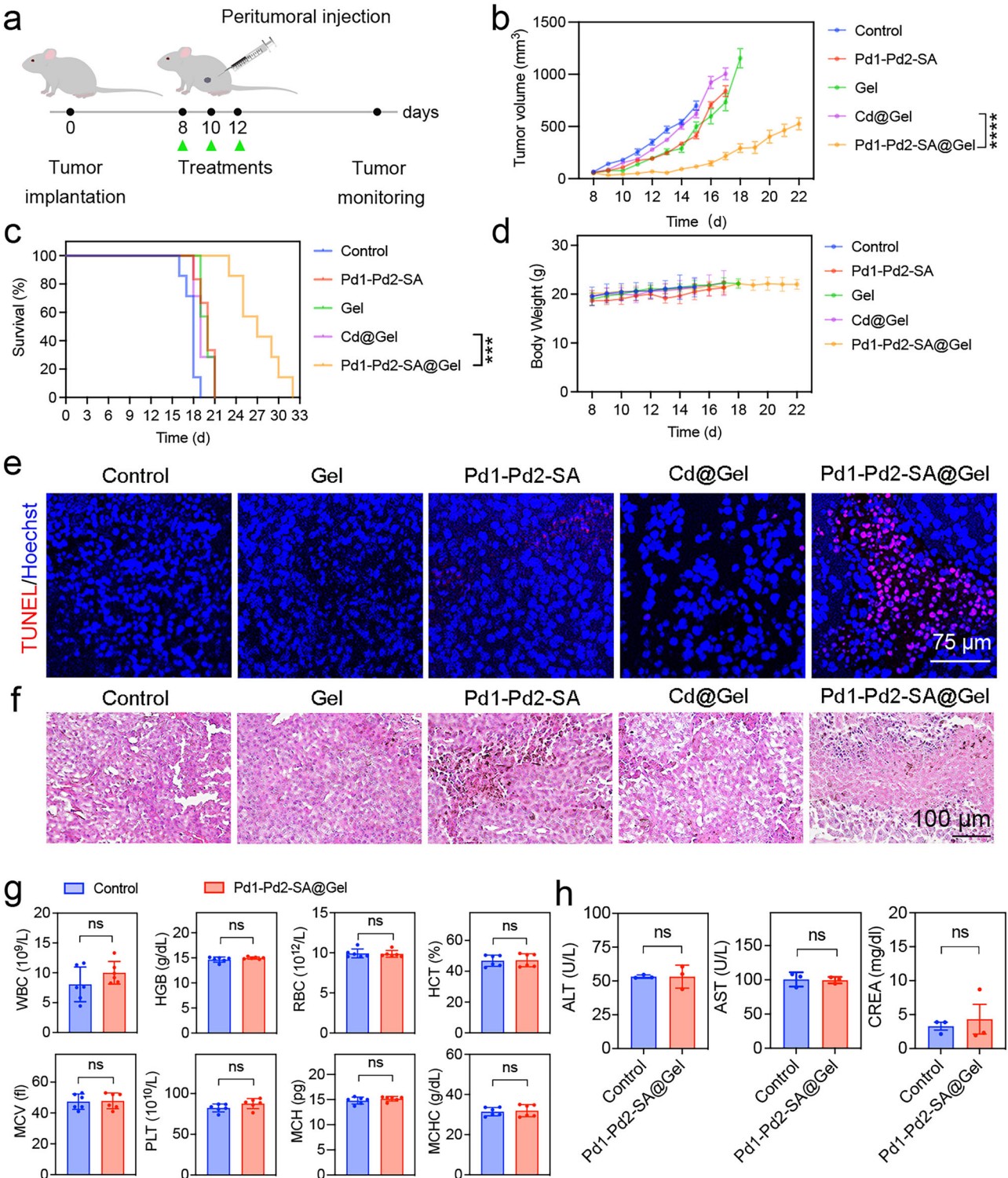

staining to evaluate the tissue damages, which showed no obvious pathological changes for all organs (Supplementary Fig. 37).

## Discussion

In summary, we have developed a TYR-based catalytic platform for treating melanin-associated skin diseases. The low barrier of the TYR-catalyzed reaction, as evidenced by the mechanistic study through the AIMD simulation, made TYR an excellent catalyst. We first validated that TYR could catalyze the in situ generation of VHL-recruiting PROTACs to efficiently degrade TYR both in vitro and in vivo for curtailing melanin synthesis to effectively treat skin hyperpigmentation. This TYR-catalyzed PROTAC design facilitates the in-situ cross-linking of two small molecules for targeted protein elimination, bypassing the need for prior covalent conjugation and mitigating issues of large molecular weights and unfavorable pharmacological properties of traditional PROTAC design. In addition, we applied this TYR-catalytic platform to activate anti-cancer prodrugs in situ through the CuAAC reaction, sensitizing drug-resistant melanoma to

**Fig. 6 | Therapeutic efficacy evaluation of TYR-activated anti-cancer prodrugs in a mouse drug-resistant melanoma model. a** Experimental schedule of TYR-catalyzed intracellular prodrug activation for treating drug-resistant B16F10 melanoma on a mouse model. **b** Tumor growth curves of DR-B16F10 melanoma-bearing mice treated with different groups (PBS, Gel, Cd@Gel and Pd1-Pd2-SA@Gel, $n = 7$ mice; Pd1-Pd2-SA, $n = 6$ mice). Data are presented as mean ± standard error (SEM). The statistical analysis was conducted between Cd@Gel and Pd1-Pd2-SA@Gel on day 15 using two-way ANOVA followed by Dunnett's multiple comparisons test. ****$P < 0.0001$. **c** Survival curves of the DR-B16F10-bearing mice treated with different treatment groups. Data are analyzed with the Log-rank (Mantel-Cox) test ($n = 7$ mice). **d** Body weight changes during the treatment course (PBS, Cd@Gel and Pd1-Pd2-SA@Gel, $n = 7$ mice; Gel, Pd1-Pd2-SA, $n = 6$ mice). Data are presented as mean ± standard deviation (SD). $P = 0.0002$. **e** TUNEL staining of the tumor tissues post-treatment. ($n = 3$ independent samples). **f** Representative images of H&E-staining of the tumor tissues post-treatment. ($n = 3$ independent samples). **g** Complete blood count analysis of the blood from mice treated with PBS and Pd1-Pd2-SA@Gel. (WBC White Blood Cell Count, HGB Hemoglobin, RBC Red Blood Cell Count, HCT Hematocrit, MCV Mean Corpuscular Volume, PLT Platelet Count, MCH Mean Corpuscular Hemoglobin, MCHC Mean Corpuscular Hemoglobin Concentration.) ($n = 6$, biologically independent mice). Data are presented as mean ± standard deviation (SD). Statistical analysis was performed *via* unpaired Student's *t*-test (two-tailed). **h** Liver (ALT Alanine Transaminase, AST Aspartate Transaminase) and kidney (CREA Creatinine) function analysis after PBS or Pd1-Pd2-SA@Gel treatments in mice. ($n = 3$ mice). Data are presented as mean ± standard deviation (SD). Statistical analysis was performed *via* unpaired Student's *t*-test (two-tailed). (ns. not significant, ***$P < 0.001$).

chemotherapeutics both in vitro and in vivo. Collectively, our de novo designed TYR-based therapeutic strategy established an integrative platform to leverage the catalytic activity of TYR to in situ generate various drugs, thereby overcoming the current treatment challenges for enhanced melanin-associated skin disease therapy.

Despite the advantages and good treatment outcomes of using in-situ TYR-catalyzed DeTYR formation as a treatment modality in treating melanin, this strategy has certain weaknesses. First, since the generation of DeTYR-3 is dependent on the presence of tyrosinase levels, how to balance the tyrosinase degradation and new production of DeTYR-3 through in-situ tyrosinase catalysis is challenging. Second, both the up- and down-regulation of tyrosinase could cause the potential side effects, including complications like Parkinson's disease. Thus, the long-term toxicity and biosafety profile of the in situ TYR-catalyzed therapeutic strategy should be carefully and comprehensively evaluated. To decrease the potential unwanted side effects, in this study, the local delivery strategy, including topical delivery for skin hyperpigmentation and peritumoral injection for skin tumors, is adopted to minimize the systemic exposure.

## Methods

### Animals
The animal study protocol was approved by the Institutional Animal Care and Use Committee (IACUC) at the University of Wisconsin-Madison. The 6–8-week-old male C57BL/6 mice and 5–6-week-old male nude mice were purchased from the Jackson Laboratory. The animals' sex was not considered in this study. The maximal tumor size allowed in the animal protocol approved by the IACUC at the UW-Madison is 1500 mm$^3$. The tumor size in this study does not exceed the maximal tumor size allowed in the protocol. Mice were housed under a 12 h light/dark cycle, an ambient temperature of 20 ± 3 °C, and 50 ± 5% humidity, with free access to water and food.

### Cell lines
The human A375 and murine B16F10 cell lines were purchased from ATCC. The primary human melanocyte (NM-2024-010-P1) was gifted by the laboratory of Dr. Vijayasaradhi Setaluri from the Department of Dermatology, the University of Wisconsin-Madison. Cells were cultured in the CO$_2$ incubator (Fisher) at 37 °C with 5% CO$_2$ and 90% relative humidity.

### mTYR-catalyzed click reaction
To determine mTYR's catalytic activity, a CuAAC reaction between pS1 and pS2 was performed. Both reactants were used at a concentration of 10 μM in H$_2$O at room temperature. The introduction of mTYR (0.1 mg/mL) induced an instantaneous blue fluorescence, confirming the enzyme's catalytic capability. The fluorescence was increased after the addition of 10 μM sodium ascorbate (SA), indicating more generation of the Fluo. SA was added to every related experiment group unless otherwise specified. Fluorescence emission was measured using a spectrometer.

### HPLC analysis of mTYR-catalyzed click reaction
The catalytic capability of mTYR and the traditional catalyst copper sulfate (CuSO$_4$) /SA was compared. Both pS1 and pS2 were used at a concentration of 10 μM in H$_2$O at room temperature. The mTYR (1 mg/mL) and CuSO$_4$ (8.35 μM) were added, respectively. The amount of Cu in CuSO$_4$ is equal to the Cu in mTYR. The concentration of SA was 41.75 μM. The total reaction time was 8 h. The targeted product, Fluo, was tested using high-performance liquid chromatography (HPLC) in the following groups: pS1 + pS2, pS1 + pS2 + mTYR, pS1 + pS2 + mTYR + SA, and pS1 + pS2 + CuSO$_4$ + SA.

### Confocal microscopy imaging
Primary human melanocyte, A375 and B16F10 cells were seeded at a density of $2 \times 10^4$ cells/well into 6-well plates for 6 h. Then, cells were treated with 10 μM of pS1 + pS2 and pS1 + pS2 + SA for 12 h. Afterward, the cells were washed thrice with PBS and stained using Wheat Germ Agglutinin 594 (WGA 594) and NucSpot® Live 488 Nuclear Stain (Nuc 488). Imaging was conducted on a Nikon AXR Confocal Microscope.

### Flow cytometry analysis
A375 and B16F10 cells were seeded at a density of $2 \times 10^5$ cells/well into 6-well plates for 6 h. Next, cells were treated with 10 μM of pS1 + pS2 and pS1 + pS2 + SA for 12 h. Afterward, cells were washed thrice with PBS. After trypsinization and resuspension in PBS, cells were subjected to flow cytometry to measure Fluo fluorescence.

### Ab initio molecular dynamics (AIMD) simulations
To study the geometric and electronic properties of TYR and its catalytic reaction between azido- and ethynyl-, AIMD simulations were performed. The simulation model containing the activating center (~400 atoms) of TYR was truncated to reduce the computational cost at a reasonable level. Models were optimized to local minima for simulations in AIMD and Metadynamics, utilizing the Grimme D3 method to compute dispersion interactions. The valence-shell electrons, defined as elements H (1s$^1$), C (2s$^2$2p$^2$), N (2s$^2$2p$^3$), O (2s$^2$2p$^4$), S (3s$^2$2p$^4$), and Cu (3d$^{10}$4s$^1$), were modeled using hybrid Gaussian and plane-wave basis sets at the triple-ζ level. A 600 Rydberg cutoff was set for the auxiliary plane wave basis sets. Geometric relaxation and Brillouin zone integration employed only the gamma point in the reciprocal space mesh. Core electrons were depicted using scalar relativistic norm-conserving pseudopotentials. The Nose-Hoover thermostat controlled the temperature in an NVT ensemble, with a simulation timestep of 1.0 fs and a total duration of up to 10 ps for system equilibration at the DFT level. A time step of 1.0 fs and a total time of a simulation run of up to 10 ps were used to equilibrate the system at the DFT level. To calculate free energy, we performed well-tempered single-walker metadynamic simulations up to 15 ps[31]. All simulations were carried out at an elevated temperature of 298 K.

### mTYR inhibition by TIn, Alk-TIn, and kojic acid
L-DOPA was used as a substrate to characterize the TYR inhibition assay. TIn, Alk-TIn, and kojic acid were dissolved in DMSO (10 mM,

100 µL) to create eight concentration gradients at a pH of 6.8. mTYR-lyophilized powder and L-tyrosine were prepared in PBS at 300 U/mL and 1 mM, respectively. Following a 20-min incubation at 37 °C, absorbance at 475 nm was recorded. An equivalent volume of DMSO-containing PBS was used as the blank control. TYR inhibition ratio was calculated by: Inhibition ratio (%) = ($\Delta A_{control}$-$\Delta A_{sample}$)/$\Delta A control \times 100$, where $\Delta A_{control}$ and $\Delta A_{sample}$ represent the absorbance change in the control and sample, respectively. The IC50 value was determined as the concentration causing 50% inhibition.

## Mechanistic study of mTYR inhibition by Alk-TIn
Inhibition kinetics of Alk-TIn were investigated at 0, 25, 50, and 100 nM, using varying concentrations of L-tyrosine (0.05, 0.1, 0.15, 0.2, 0.25, 0.3, 0.35, and 0.4 mM). Analysis was conducted via Lineweaver-Burk plots to determine the inhibition model, employing the same protocol used for IC50 evaluations. Measurements were done in triplicate.

## In silico modeling of DeTYR-3 ternary complex
Protein-protein docking was performed using the Rosetta software suite. Conformer generation was conducted with RDKit, an open-source cheminformatics tool, version 2020.03.1. Detailed procedures were described in the Supporting Information. Additionally, Python was employed to develop computational tools for PROTAC ternary model construction.

## Cell viability assay
Human A375 and murine B16F10 cells were cultured in DMEM, 10% fetal bovine serum (FBS), and 1% penicillin-streptomycin. A375 and B16F10 cells were seeded in a 96-well plate at a density of 5000 cells/well and incubated for 24 h. The cells were then exposed to incremental concentrations (0 to 50 µM) of VH032 + Alk-TIn, VH032-Azi1 + Alk-TIn + SA, VH032-Azi2 + Alk-TIn + SA, and VH032-Azi3 + Alk-TIn + SA for 24 h. Afterward, the medium was replaced with CCK-8 solution, and the plates were incubated at 37 °C for 1 or 1.5 h. Absorbance at 450 nm was determined with a microplate reader.

## TYR inhibition by VH032-Azi3 + Alk-TIn + SA in cells
A375 and B16F10 cells, seeded at a density of $3 \times 10^4$ cells/well in 24-well plates, were incubated at 37 °C for 24 h. After the addition of 1 µM α-MSH, treatment groups were exposed to 0.1 µM of kojic acid, TIn, Alk-TIn, and VH032-Azi3 + Alk-TIn + SA, whereas the control group only received DMSO-containing medium (<0.3%). After a 24 h incubation, cells were collected, rinsed with PBS, and lysed using 1% Triton X-100 (100 µM in 50 mM PBS, pH 6.8) with 0.1 mM phenylmethylsulfonyl fluoride (PMSF). Cell lysates were subjected to freeze-thaw cycling and centrifuged at 13,800 g for 15 min at 4 °C. Afterward, the supernatant was assayed. The supernatant (80 µL) was analyzed in a 96-well plate by adding 20 µL of 2 mM L-dopa and incubating at 37 °C for 1 h. Absorbance at 475 nm was quantified every 10 min. % TYR activity = $OD_{experimental\ group}$/$OD_{control\ group} \times 100$.

## Melanin content evaluation
A375 and B16F10 cells were seeded at a density of $3 \times 10^4$ cells/well into 24-well plates, followed by a 24 h incubation at 37 °C. Treatments included 1 µM of α-MSH and 0.1 µM of kojic acid as positive controls, and 0.1 µM of TIn, Alk-TIn, and VH032-Azi3 + Alk-TIn + SA as experimental groups. After a 24 h incubation, cells were harvested, washed with PBS, lysed in 1 M NaOH, and heated at 60 °C for 1 h. The absorbance of supernatants was recorded at 405 nm. Melanin content (%) = $OD_{experimental\ group}$/$OD_{control\ group} \times 100$.

## Western blot analysis
A375 and B16F10 cells were seeded at a density of $2 \times 10^5$ cells/well in DMEM with 10% FBS. After a 24 h incubation, cells were then exposed to different concentrations of compounds or DMSO-containing medium for 12 and 24 h in 6-well plates. Following treatments, cells underwent washing with ice-cold PBS, harvesting in RIPA lysis buffer, being lysed on ice for 30 min, and being centrifuged for 30 min at 16,200 g. The protein contents in the supernatants were quantified using a bicinchoninic acid (BCA) assay. Following protein extraction, supernatants were prepared with a loading buffer and heated at 95 °C for 15 min. Each sample (15 µg of protein) was loaded into a 12% SDS-polyacrylamide gel for protein separation. Afterward, proteins were transferred to PVDF membranes, blocked with 5% nonfat milk in $H_2O$, and incubated with primary antibodies against MITF and TYR (1:1000) at 4 °C overnight. Followed by a 1 h room temperature incubation with goat anti-rabbit IgG H&L (HRP) secondary antibodies (1:10,000). Protein bands were visualized using electrochemiluminescence (ECL) and quantified with ImageJ software.

## Co-Immunoprecipitation assay
A375 and B16F10 cells were seeded at a density of $2 \times 10^5$ cells/well in DMEM containing 10% FBS. After a 24 h incubation, the medium in the corresponding flasks, except the control, was replaced with medium containing VH032-Azi3 (0.1 µM) + Alk-TIn (0.1 µM) + SA (0.5 µM) or epi-VH032-Azi3 (0.1 µM) + Alk-TIn (0.1 µM) + SA (0.5 µM). Following a 4 h incubation, cells were rinsed with cold PBS and subsequently lysed on ice in IP lysis buffer supplemented with 1 mM PMSF and phosphatase inhibitors for 30 min. Subsequently, lysates were centrifuged at 16,200 g for 10 min at 4 °C. Afterward, 10 µL of lysate supernatants were saved as the input samples. 25 µL of magnetic beads was added to each supernatant in microcentrifuge tubes, and the resulting mixtures were incubated at 4 °C, and mixed gently for 30 min. Next, the sample-containing tube was placed on a magnetic rack. The precleared samples were transferred to a new chilled tube. The total proteins in the precleared samples were quantified through the BCA assay. For each assay, 10 µg of anti-VHL antibody (Santa Cruz Biotechnology) was added to the precleared lysate and incubated for 2 h at room temperature with rotation. The antigen-antibody mixture was then transferred to pre-washed magnetic beads and incubated for an additional hour at room temperature with mixing. The beads were separated magnetically, retaining the flow-through for analysis. After three washes with 500 µL of Wash Buffer and a final rinse with 500 µL of purified water, the beads were processed for Western blot analysis by resuspending in WB loading buffer, followed by boiling and examination for TYR or VHL proteins. The primary antibodies against TYR and VHL were diluted to 1:1000.

## Immunocytochemistry analysis of TYR degradation in living cells
A375 and B16F10 cells were plated in confocal dishes for 24 h. VH032 (0.1 µM), Alk-TIn (0.1 µM), VH032 (0.1 µM) + Alk-TIn (0.1 µM), VH032-Azi1 (0.1 µM) + Alk-TIn (0.1 µM) + SA (0.5 µM), VH032-Azi2 (0.1 µM) + Alk-TIn (0.1 µM) + SA (0.5 µM), and VH032-Azi3 (0.1 µM) + Alk-TIn (0.1 µM) + SA (0.5 µM) were added and incubated with the cells for 12 h, respectively. Afterward, cells were washed thrice with cold PBS and then fixed in 4% paraformaldehyde in PBS (pH 7.4) at room temperature for 10 min, followed by three washes with cold PBS. Permeabilization was achieved with 0.2% Triton X-100 in PBS for 10 min. To block nonspecific antibody binding, samples were incubated with 1% BSA in PBST (0.1% Tween 20 in PBS) for 30 min. Primary antibodies, diluted in 1% BSA in PBST, were applied for 1 h at room temperature in a humidified chamber, followed by three PBS washes. Afterward, cells were stained with a secondary antibody (containing 1% BSA) at room temperature for 1 h under dark conditions. Subsequently, cells were stained with 0.1 µg/mL Hoechst for 1 min, followed by three cold PBS washes. Coverslips were mounted with mounting medium and sealed. Fluorescence imaging was conducted on a Nikon AXR Confocal Microscope.

## LC-MS analysis of intracellular DeTYR-3

A375 and B16F10 cells were seeded in 6-well plates and treated with VH032-Azi3 (1 μM) + Alk-TIn (1 μM) + SA (5 μM) for 24 h. Cells were washed with PBS, trypsinized, and resuspended in ultrapure water. Following sonication for lysis, the cell lysates were centrifuged for 5 min at 13,800 g. Supernatants were treated with cold acetonitrile, stored at −20 °C for 1 h, and centrifuged at 13,800 g for 20 min to obtain the protein precipitate, which was then solubilized in methanol and subjected to LC-MS.

## Cream preparation

For the control cream, cetyl esters wax (6.25 g) and white wax (2.8 g) were initially melted, followed by the addition of 2.8 g of mineral oil. Afterward, the mixture was heated to 70 °C. Simultaneously, 25 mg sodium borate was dissolved in purified water heated to the same temperature. This solution was incrementally mixed into the oleaginous base. Other formulations were prepared with 0.03% (w/w) of various compounds (final concentration: 1 mM). The blend was then cooled with continuous rapid stirring until it solidified.

## In vivo treatment of skin hyperpigmentation

Male nude mice (5–6 weeks) were used to establish a skin hyperpigmentation model. An ethanol solution of 15% DHA (w/w) and 3% erythrulose (w/w) was applied to the backs of mice in a square pattern, followed by a UV irradiation (300 μW/cm² at 10 W, Farmingdale, NY, USA) for 15 min. Three days later, these mice were randomly assigned into seven groups ($n = 6$ per group) for subsequent treatments, including (1) blank control; (2) VH032; (3) Alk-TIn; (4) VH032 + Alk-TIn; (5) epi-VH032-Azi3 + Alk-TIn + SA, (6) DeTYR-3 and (7) VH032-Azi3 + Alk-TIn + SA creams at concentrations of 0.03% (w/w). After treatments, mice were euthanized, and pigmented skin samples were harvested. These samples were fixed in 10% neutral buffered formalin, embedded in paraffin, sectioned to 5 μm, and stained using H&E. A Nikon Intensilight fluorescence microscope was utilized for examination. Melanin quantification was assessed with a VitroView™ Fontana-Masson Stain Kit, following manufacturer guidelines. Positive cell proportions were determined by comparing the number of positive to total epidermal cells in three distinct images per sample.

## Western blot analysis of skin tissues

After finishing seven groups of treatments for skin hyperpigmentation, mice were euthanized at 48 h, and pigmented skin samples were harvested on ice. Tissue samples were transferred to round-bottomed microcentrifuge tubes and immediately snap-frozen in liquid nitrogen. The skin samples were mechanically processed to a single-cell solution under collagenase digestion (1 mg/mL). After centrifugation, the skin samples were lysed, and the western blot assay was performed.

## Quantification of melanin contents in skin tissues

Skin samples were digested in 2.5 M NaOH for 24 h, then added to 1 M NaOH with 10% DMSO (v/v) at a concentration of 50 mg/mL, and incubated at 80 °C for 1 h. After centrifugation to remove debris, the melanin content was measured by detecting the absorbance at 405 nm.

## Generation of drug-resistant cancer cells

Drug-resistant A375 and B16F10 (DR-A375 and DR-B16F10) cell lines were generated through the treatment with a gradual increase in the dosage of the cancer drug (Cd) from an initial 20 μM to a final 100 μM until the cancer cells could well tolerate the presence of 100 μM Cd. Subsequently, the viability of DR and parental cell lines was evaluated using CCK-8 assays at Cd concentrations ranging from 0.0001 to 10000 μM. The Resistance Index, defined as the ratio of the IC50 (drug concentration causing 50% growth inhibition) values of resistant to parental cells, was used to quantify resistance capability.

$$\text{Resistance Index} = IC_{50} \text{of resistant cells}/IC_{50} \text{of parental cells}.$$

## Viability assay of DR cancer cells

DR-A375 and DR-B16F10 cells were cultured in 96-well plates at a density of 5000 cells/well and incubated for 24 h. Subsequently, Pd1 and Pd2 (0, 3.125, 6.25, 12.5, 25, 50, and 100 μM) were added. Following a further 24 h incubation, the medium was replaced with the CCK-8 solution, and the cells were incubated at 37 °C for 1 to 1.5 h. Absorbance was then recorded at 450 nm using an Infinite M Plex microplate reader.

## HPLC analysis of intracellular and extracellular drug content

To study the drug-resistant mechanism of Cd and Pd1 + Pd2 + SA, both intracellular and extracellular relative drug content have been tested by high performance liquid chromatography (HPLC). The DR-A375 and DR-B16F10 cells were plated in 6-well plates. The Cd and Pd1 + Pd2 + SA at indicated concentrations (100 μM) were added, respectively. After 24 h and 48 h incubations, the supernatants of DR-A375 and DR-B16F10 cells were collected. After being washed with PBS 2 times, the cells were harvested using trypsin and resuspended in PBS. After lysis, the intracellular and extracellular Cd were analyzed using HPLC.

## Flow cytometry analysis of prodrug activation-induced apoptosis

DR-A375 and DR-B16F10 cells, seeded at a density of $2 \times 10^5$ cells/well in 6-well plates, were incubated for 12 h before treatment with 100 μM of Pd1, Pd2, Cd, Pd1 + Pd2, and Pd1 + Pd2 + SA for 24 h. Following the treatment phase, cells underwent staining with Annexin V (5 μL) in 100 μL of binding buffer for 15 min at room temperature in the dark. Propidium iodide (PI) (1 μL) in 100 μL of binding buffer was added for 15 min without rinsing. Flow cytometry was employed to assess Annexin V-PI staining, using fluorescein isothiocyanate-Annexin V (Ex = 488 nm, Em = 520/21 nm) and PI (Ex = 561, Em = 590/30 nm).

## 3D tumor sphere inhibition

1.5% agarose gel was prepared by mixing 6 mL DMEM and 90 mg agarose in an 80 °C water bath for 30 min. Afterward, the agarose gel was added to each well of a 96-well plate for gelation. DR-A375 and DR-B16F10 cells were seeded into each well, and the tumor sphere was monitored every day to grow to a certain size. Afterward, the tumor spheroids with a similar size and shape were selected to be treated with Cd and the mixture of Pd1 + Pd2 + SA (0, 3.125, 6.25, 12.5, 25, 50, and 100 μM). Following 48 h of incubation, the medium was discarded, and the growth and morphology of the 3D tumor spheres were examined using a Nikon Intensilight Fluorescence Microscope.

## Hydrogel preparation

The Pluronic F127 hydrogel was prepared by dissolving Pluronic F127 (2.1 g) in 10 mL PBS at ambient temperature. The gelation time was assessed by incubating at 37 °C.

## In vivo anti-tumor efficacy of Pd1-Pd2-SA@Gel

The male C57BL/6 mice (6–8 weeks) were subcutaneously implanted with DR-B16F10 cells ($1 \times 10^6$ cells in 100 μL PBS). Afterward, mice bearing drug-resistant B16F10 tumors (50–100 mm³) were divided into five groups (PBS, Gel, Cd@Gel, and Pd1-Pd2-SA@Gel, $n = 7$; Pd1-Pd2-SA, $n = 6$). Treatments were administered peritumorally at a dosage of 5 mg/kg every two days three times in total. Body weight and tumor volume were monitored daily. Mice were euthanized by cervical dislocation when tumor sizes >1500 mm³. Survival time was recorded. Tissues from tumors and major organs were preserved in 4% paraformaldehyde for histological examination, including H&E and TUNEL

staining, to investigate the anti-tumor mechanisms. Furthermore, blood was drawn to assess hematological (WBC, RBC, HGB, HCT, MCV, MCH, MCHC, PLT) ($n = 6$ each), liver function (ALT and AST) ($n = 3$ each), and kidney function markers (CREA) ($n = 3$ each).

## Statistical analysis

Data throughout the experiment are presented as mean ± standard deviation (SD) or mean ± standard error (SEM), analyzed using GraphPad Prism. Two-group comparisons were analyzed with the unpaired Student's $t$-test. Analysis of variance (ANOVA) was used to compare multiple groups (>two groups) followed by Dunnett's multiple comparisons test.

## Reporting summary

Further information on research design is available in the Nature Portfolio Reporting Summary linked to this article.

## Data availability

The authors provide all supporting data for this study in the article and the Supporting Information. Source data are provided with this paper.

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

## Acknowledgements

We want to thank the optical imaging core, small animal facilities, flow cytometry core, histological core, and UW-Madison Wisconsin Centers for Nanoscale Technology for their help with this study. This work was supported, in part, by METAVIVOR Foundation Early Career Research Grant Award, the American Cancer Society Research Scholar Grant (Grant number: RSG-23-1140821-01-ET, to Q.H.), and the V Foundation Scholar Grant (to Q.H.). This work was also partially supported by NIH grants R01EB035992 (to Q.H.) and R01CA288851 (to Q.H.). The content is solely the responsibility of the authors and does not necessarily represent the official views of the National Institutes of Health. We also thank the support from the University of Wisconsin Carbone Cancer Center Research Collaborative and the Pancreas Cancer Task Force, and the start-up package from the University of Wisconsin-Madison.

## Author contributions

Y.Y. and Q.H. conceived the project. Y.Y., Z.G., Y.W., and S.Y. carried out experiments and interpreted data. Y.Y. and Q.H. wrote the manuscript with input from all authors. Q.H. provided supervision.

## Competing interests

The authors declare no competing interests.
