## [Transparent Peer Review file · Nature Communications]

Endogenous Tyrosinase-Catalyzed Therapeutics

Corresponding Author: Dr Quanyin Hu

Version 0:

Reviewer comments:

Reviewer #1

(Remarks to the Author)

Abnormal accumulation or reduction of melanin could be associated with skin and eye diseases. Tyrosinase is a central enzyme in the melanin production pathway, which could be a potential target in developing therapeutics to treat melanin-associated disorders. In this work, the authors demonstrated that formed proteolysis-targeting chimeras (PROTACs) degraded intracellular tyrosinase (TYR) protein to decrease melanin synthesis for treating hyperpigmentation. For this purpose, the authors developed a TYR-based in situ catalytic system that can generate drugs intracellularly through an endogenous copper-catalyzed azide-alkyne cycloaddition (CuAAC) reaction and formed proteolysis-targeting chimeras (PROTACs) to degraded intracellular TYR protein. It is a very interesting work, and the authors performed a lot of experiments to support their claims. Moreover, they demonstrate in a mouse model that tyrosinase-catalyzed therapeutics could efficiently change skin hyperpigmentation within 48 h. The success of the experiment suggests the potential for a melanoma treatment.

In my opinion, this work is missing the discussion of the potential side effects of the treatment in humans associated with the disruption of normal function of tyrosinase. Loss of tyrosinase protein caused by the treatment could change the color of the skin, and the normal vision of the patient. Skin, hair, and eye color changes associated with tyrosinase degradation could cause albinism-associated defects. Moreover, degradation of tyrosinase could disturb the normal function of this protein and potentially cause Parkinson's disease. All these complications might limit or make it not possible the use the technology for melanoma treatment in the future.

Another question is associated with the implications of computational biology. The paper has demonstrated a deep knowledge of molecular modeling. The catalytic activity of tyrosinase is mechanistically validated by ab initio molecular dynamics theoretical calculation and experimental catalysis performance. To study the geometric and electronic properties of tyrosinase and its catalytical reaction molecular dynamics simulations were performed for the truncated activating center (~400 atoms), which has been selected from the mushroom tyrosinase atomic structure. Unfortunately, the authors do not provide any detailed information on which amino acid residues were included in the activating center. This initial simplification using the truncated center significantly decreases the validity of computer simulations and computational results. As an alternative, authors could use the AlphaFold atomic structure of mice tyrosinase, which is available from the UniProt database.

Reviewer #2

(Remarks to the Author)

The work reports the application of a TYR catalyzed click reaction of azide with alkyne in anti-cancer drug discovery. The discovery of copper centered catalytic system with TYR is a new, interesting without external copper catalyst, making biocompatible. The studies are carried out both in vitro and in vivo models. The results are impressive. I recommend it to Nat Commun after addressing the following issues:

It is claimed (Fig 1) that "Mechanistically, this increase in the catalytic efficacy was attributed to enhanced conversion from met-tyrosinase to deoxy-tyrosinase by SA." Pls provide evidence.

In an enzyme catalyzed reaction, substrate(s) should favorably bind to catalytic sites. No rationales were provided for the azide and the alkynyl reactants although the ab initio molecular dynamics studies indicated with the azide and the alkynyl groups in both substrates could bind to the catalytic sites of TYR.

The studies did not explain why SA could facilitate the reaction. Why was SA not used in the PROTACs, but required in the prodrugs?

In the in vivo PROTAC anti-tumor experiments, the preformed DeTYR-3 degrader should be included for comparison studies.

Reviewer #3

(Remarks to the Author)

In this manuscript, Hu et al. developed endogenous tyrosinase-catalyzed therapeutics for treating melanin-associated skin diseases and drug-resistant tumors. Through the catalytic activity of copper-containing tyrosinase, a series of drug precursors can undergo click reactions in situ to generate effective PROTAC molecules that control disease progression. This is an innovative approach which has been validated to be very effective. Overall, this is an excellent piece of work, and the manuscript is well written. The authors may consider the following comments for revisions.

1. The supporting information do not specify the amount of traditional catalysts, such as copper sulfate/SA. Are their added quantities reasonable, and are they equimolar to mTYR? Beyond fluorescence data, the authors should also provide HPLC evidence to demonstrate that mTYR exhibits superior catalytic efficiency for target molecules compared to traditional catalysts. In Fig. 2, the authors provided the mechanistic studies of TYR's catalytic activity. The advantages of such enzyme-catalyzed reaction and efficiency compared to traditional copper catalysts should be discussed briefly in this section.

2. The confocal imaging data in Fig. 1d show that the fluorescence of "Fluo" is distributed throughout the entire cell, including the nucleus. Why does "Fluo" fluorescence appear in the nucleus considering that based on the data provided in Supplementary Fig. 10, TYR is distributed only in the cytoplasm? The authors should provide a reasonable explanation. Selecting an appropriate imaging time point to achieve colocalization of "Fluo" fluorescence with TYR distribution would make the catalytic role of TYR in generating fluorescence more convincing.

3. Will TYR-catalyzed TYR degraders have adverse effects on normal cells? I am curious whether the "Fluo" generation shown in Figure 1d would be higher in the B16F10 cell line compared to normal cell lines. Additionally, the TYR content in skin-related normal cell lines and TYR-overexpressing cell lines (B16F10 and A375) should be quantitatively characterized using flow cytometry or ELISA.

4. How long can the efficacy of TYR-catalyzed TYR degraders be sustained? While PROTAC molecules can continuously degrade TYR in a cyclic manner, this also means the rate of catalytic production of new effective PROTAC molecules will gradually decrease. How to achieve a balance between these two processes? In Fig. 4a, the authors only observed the therapeutic effects over six days. After TYR levels reach a low point, the production of PROTAC molecules may no longer be effective. Could this lead to a recurrence of hyperpigmentation over time? In addition, why does the use of a cream formulation enable effective penetration of TYR degrader precursors into the skin? The authors should use other methods (e.g., fluorescence labeling) to verify that the drug molecules indeed penetrate the skin.

5. When applied in vivo, particularly for tumor treatment involving the generation of toxic drugs, how can the authors ensure that the click reaction is specifically catalyzed by tumor or disease cells with high TYR expression, rather than by other copper-containing proteins in the blood?

6. Some sections of the manuscript should include appropriate references to enhance the credibility. For example, the statement: "Recent developments in CuAAC have shifted towards endogenous Cu catalysts, which catalyze chemical reactions without adding exogenous Cu, mitigating the risk of perturbing copper homeostasis and toxicity against normal tissues".

Version 1:

Reviewer comments:

Reviewer #1

(Remarks to the Author)

[Editorial note: Reviewer #1 only provided confidential comments but they are satisfied with revisions and have no further concerns.]

Reviewer #2

(Remarks to the Author)

The authors did a good job on addressing the issues raised by reviewers including mine. I recommend it for Nat Commun

without further action.

Reviewer #3

(Remarks to the Author)

The authors have conducted additional experiments, revised the results and discussion and addressed all the questions. It is ready to be published.

We thank the reviewers for the valuable feedback on our manuscript. We have carefully revised the manuscript according to the reviewers' comments. The point-by-point responses are provided as follows.

Response to Reviewer #1

Abnormal accumulation or reduction of melanin could be associated with skin and eye diseases. Tyrosinase is a central enzyme in the melanin production pathway, which could be a potential target in developing therapeutics to treat melanin-associated disorders. In this work, the authors demonstrated that formed proteolysis-targeting chimeras (PROTACs) degraded intracellular tyrosinase (TYR) protein to decrease melanin synthesis for treating hyperpigmentation. For this purpose, the authors developed a TYR-based in situ catalytic system that can generate drugs intracellularly through an endogenous copper-catalyzed azide-alkyne cycloaddition (CuAAC) reaction and formed proteolysis-targeting chimeras (PROTACs) to degraded intracellular TYR protein. It is a very interesting work, and the authors performed a lot of experiments to support their claims. Moreover, they demonstrate in a mouse model that tyrosinase-catalyzed therapeutics could efficiently change skin hyperpigmentation within 48 h. The success of the experiment suggests the potential for a melanoma treatment.

1. In my opinion, this work is missing the discussion of the potential side effects of the treatment in humans associated with the disruption of normal function of tyrosinase. Loss of tyrosinase protein caused by the treatment could change the color of the skin, and the normal vision of the patient. Skin, hair, and eye color changes associated with tyrosinase degradation could cause albinism-associated defects. Moreover, degradation of tyrosinase could disturb the normal function of this protein and potentially cause Parkinson's disease. All these complications might limit or make it not possible the use the technology for melanoma treatment in the future.

Response: We appreciated the reviewer's insightful comments. We agree with the reviewer that there is an associated potential side effects with down- or up-regulation of tyrosinase, and we should be very careful when applying therapeutics that target/degrade intracellular tyrosinase for disease treatment. With these potential side effects in mind, in this study, we selected the local delivery strategy rather than systemic administration routes (e.g., intravenous injection) to treat the skin hyperpigmentation and drug-resistant melanoma. Specifically, the creams are applied to the backs of mice in a specific square pattern for the pigmentation model. Therefore, DeTYR-3 is, in principle, formed in situ by overexpressed TYR in pigmented melanocytes. For the drug-resistant melanoma model, we used peritumoral injections to maximize the exposure of prodrugs to the tumor only. This peritumoral injection for skin tumors offers advantages over a conventional intravenous administration strategy by delivering therapeutics directly around the tumor, offering higher tumor-localized drug concentrations and reduced systemic side effects. To further minimize the reviewer's concern, we performed the long-term efficacy and toxicity of the DeTYR-3-based treatment strategy against skin hyperpigmentation. As shown in **Fig. R1&2**, this in-situ formed DeTYR-3 treatment maintained a long-term therapeutic outcome with no indication of recurrence

of skin hyperpigmentation over an evaluation course of 30 days. Besides, we did not find any abnormal changes in skin color or adverse damage to skin cells or infiltration of immune cells, as shown in H&E staining of skin. One potential explanation of these observations is that the DeTYR-3 generation in this design is dependent on the presence of TYR intracellularly. Even though the balance of the TYR degradation and DeTYR-3 generation is very challenging to achieve in terms of maximizing the efficacy and minimizing toxicity. Theoretically, more TYR degradations will induce less DeTYR-3 production, which will weaken the efficacy of DeTYR-3. However, this dependence of TYR and DeTYR-3 generation, to some extent, ensures the safety profile of this strategy since the low intracellular TYR level will not generate more DeTYR-3.

We have added the discussion in the revised manuscript to highlight the importance of considering the potential side effects of this treatment strategy, as shown below.

“Despite the advantages and superior treatment outcomes of using in-situ TYR-catalyzed DeTRY formation as a novel treatment modality in treating melanin, this strategy has certain weaknesses. First, since the generation of DeTYR-3 is dependent on the presence of tyrosinase levels, how to balance the tyrosinase degradation and new production of DeTYR-3 through in-situ tyrosinase catalysis is challenging. Second, both the up- and down-regulation of tyrosinase could cause the potential side effects, including complications like Parkinson's disease. Thus, the long-term toxicity and biosafety profile of the in situ TYR-catalyzed therapeutic strategy should be carefully and comprehensively evaluated. To decrease the potential unwanted side effects, in this study, the local delivery strategy, including topical delivery for skin hyperpigmentation and peritumoral injection for skin tumors, is adopted to minimize the systemic exposure.”

Fig. R1 (Supplementary Fig. 28 in Supplementary Information). Images of square pigmentation patterns on the backs of mice treated with blank control and VH032-Azi3 + Alk-TIn + SA creams at different time points.

Fig. R2. H&E staining of mouse skin tissues treated with blank control and VH032-Azi3 + Alk-TIn + SA creams on day 30.

2. Another question is associated with the implications of computational biology. The paper has demonstrated a deep knowledge of molecular modeling. The catalytic activity of tyrosinase is mechanistically validated by ab initio molecular dynamics theoretical calculation and experimental catalysis performance. To study the geometric and electronic properties of tyrosinase and its catalytical reaction molecular dynamics simulations were performed for the truncated activating center (~400 atoms), which has been selected from the mushroom tyrosinase atomic structure. Unfortunately, the authors do not provide any detailed information on which amino acid residues were included in the activating center. This initial simplification using the truncated center significantly decreases the validity of computer simulations and computational results. As an alternative, authors could use the AlphaFold atomic structure of mouse tyrosinase, which is available from the UniProt database.

Response: Thanks for the reviewer's kind suggestion. According to the reviewer's suggestion, we have changed the structure of mushroom tyrosinase to the AlphaFold atomic structure of mouse tyrosinase for the whole docking experiment. We have added a description of the amino acid residues included in the activating center in our revised manuscript. The details are shown below (**Fig. R3**). "In the activation center of two copper ions, there are six histidine amino acid residues around the active position, namely His367, His363, His390, His211, His202, and His180."

Fig. R3 (Fig 1a in the main text). Six histidine amino acid residues in the activation center of two copper ions in the AlphaFold atomic structure of mouse tyrosinase are His367, His363, His390, His211, His202, and His180.

Response to Reviewer #2

The work reports the application of a TYR catalyzed click reaction of azide with alkyne in anti-cancer drug discovery. The discovery of copper centered catalytic system with TYR is a new, interesting without external copper catalyst, making biocompatible. The studies are carried out both in vitro and in vivo models. The results are impressive. I recommend it to Nat Commun after addressing the following issues.

1. It is claimed (Fig 1) that “Mechanistically, this increase in the catalytic efficacy was attributed to enhanced conversion from met-tyrosinase to deoxy-tyrosinase by SA.” Pls provide evidence.

Response: Thanks for your comments. We have added further explanations and cited supporting references in our revised manuscript. Please see below.

“Tyrosinase catalysis begins with met-tyrosinase binding to o-diphenol, leading to the formation of o-quinone and the reduction of the enzyme to deoxy-tyrosinase.^{1, 2} Sodium ascorbate (SA) reduces the resulting o-quinone back to o-diphenol, effectively regenerating the substrate and facilitating the transition of met-tyrosinase to deoxy-tyrosinase.^{3, 4} The copper centers in met- and deoxy-tyrosinase exist in Cu (II) and Cu (I) states, respectively.⁵ Accordingly, SA enhances catalytic performance in Cu (I)-driven click reactions by stabilizing the active Cu (I) state.”

References:

1. Kampmann, M., Riedel, N., Mo, Y.L., Beckers, L. & Wichmann, R. Tyrosinase catalyzed production of 3,4-dihydroxyphenylacetic acid using immobilized mushroom (*Agaricus bisporus*) cells and in situ adsorption. *J. Mol. Catal. B:Enzym.* **123**, 113-121 (2016).
2. Zolghadri, S. & Saboury, A.A. in *The Enzymes*, Vol. 56. (ed. C.T. Supuran) 31-54 (Academic Press, 2024).
3. Yin, X. et al. Chemical Stability of Ascorbic Acid Integrated into Commercial Products: A Review on Bioactivity and Delivery Technology. *Antioxidants* **11**, 153 (2022).
4. Weerawardana, M.B.S., Thiripuranathar, G. & Paranagama, P.A. Natural Antibrowning Agents against Polyphenol Oxidase Activity in *Annona muricata* and *Musa acuminata*. *J. Chem.* **2020**, 1904798 (2020).
5. Mertens, B.S., Moore, M.D., Jaykus, L.-A. & Velev, O.D. Efficacy and Mechanisms of Copper Ion-Catalyzed Inactivation of Human Norovirus. *ACS Infect. Dis.* **8**, 855-864 (2022).

2. In an enzyme catalyzed reaction, substrate(s) should favorably bind to catalytic sites. No rationales were provided for the azide and the alkynyl reactants although the ab initio molecular dynamics studies indicated with the azide and the alkynyl groups in both substrates could bind to the catalytic sites of TYR.

Response: Thanks for raising this question. Alk-TIn has the TYR targeting capability and

selectively binds to TYR since it uses the TYR selective inhibitor as the warhead. However, there is no such targetability of VHL-Azi towards TYR. We hypothesize that it will transport intracellularly in free form. To further understand the binding sites of Alk-TIn and VH032-Azi, we replaced the mushroom TYR (mTYR) in the original manuscript with the AlphaFold atomic structure of mouse TYR and re-did the protein-protein docking.

Based on Rosetta protein-protein docking protocol, the decoy with the distance between VH032-Azi3 and Alk-TIn atoms was less than 21 Å (**Fig. R4a**), occupying significant cavities within the VHL E3 ligase and TYR. Hydrogen bonding was observed between Alk-TIn and residues H367, H363, H390, A201 and between VH032-Azi3 and residues Y98, H115, H110, and D197. Importantly, neither the azido nor ethynyl groups interacted with residues, affirming the feasibility of the click reaction. The docking profile identified the most stable VHL-DeTYR-3-TYR ternary complex (**Fig. R4b**). Specifically, the Alk-TIn moiety bound to H363 and V377 of TYR, and the VH032-Azi3 moiety bound to Y98 and Q376 of VHL, both of which were through hydrogen bonds. These interactions and conformations suggest that VH032-Azi3 and Alk-TIn could form a stable ternary complex with VHL and TYR.

Fig. R4 (Fig 3c, d in the main text). Mechanism of Action (MoA) exploration of TYR-catalyzed in-situ formed TYR degraders. a, Molecular dynamic simulation of TYR (wheat), Alk-TIn (yellow), VHL (cyan), and VH032-Azi3 (magenta). The distance was measured by the two linker atoms (circled in green). **b**, Molecular dynamic simulation of DeTYR-3 (yellow) interacting with VHL (cyan) and TYR (wheat) proteins.

3. The studies did not explain why SA could facilitate the reaction. Why was SA not used in the PROTACs, but required in the prodrugs?

Response: Thanks for the comments. To further clarify the critical role of SA, we added the following statements in the revised manuscript.

“Sodium ascorbate (SA) can reduce o-quinones, which are the products of tyrosinase-catalyzed reactions, back to o-diphenols.^{3,4} This reduction helps to regenerate the substrate for tyrosinase, promoting the conversion of met-tyrosinase to deoxy-tyrosinase. The valence state of copper in met-tyrosinase and deoxy-tyrosinase is divalent (Cu (II)) and monovalent (Cu (I)), respectively. Therefore, the catalytic efficacy was enhanced by SA in Cu (I)-catalyzed click reaction.”

For the second question, the SA was actually used in both PROTACs and anti-cancer prodrugs. To avoid any misunderstanding, we described it in the caption of **Figure 3** and added SA in **Figures 3b, 3e, and 3f-3i** in the revised manuscript, as shown below (**Fig. R5**).

Fig. R5 (Fig. 3 in the main text). TYR-catalyzed in-situ formed TYR degraders. **a**, Chemical structure of azido-modified VH032 ligands (VH032-Azi1, VH032-Azi2, VH032-Azi3); alkyne-

modified TYR inhibitor (Alk-TIn), and click-catalyzed TYR degraders (DeTYR-1, DeTYR-2, DeTYR-3). **b**, Western blot analysis of TYR and MITF in A375 and B16F10 cells treated with VH032, Alk-TIn, VH032 + Alk-TIn, VH032-Azi1 + Alk-TIn + SA (0.5 μ M), VH032-Azi2 + Alk-TIn + SA (0.5 μ M) and VH032-Azi3 + Alk-TIn + SA (0.5 μ M) at the concentration of 0.1 μ M for 24 h. **c**, Molecular dynamic simulation of TYR (wheat), Alk-TIn (yellow), VHL (cyan), and VH032-Azi3 (magenta). The distance was measured by the two linker atoms (circled in green). **d**, Molecular dynamic simulation of DeTYR-3 (yellow) interacting with VHL (cyan) and TYR (wheat) proteins. **e**, Co-immunoprecipitation of TYR and VHL from lysates of A375 and B16F10 cells treated with VH032-Azi3 (0.1 μ M) + Alk-TIn (0.1 μ M) + SA (0.5 μ M) or epi-VH032-Azi3 (0.1 μ M) + Alk-TIn (0.1 μ M) + SA (0.5 μ M) for 4 h. **f, g**, TYR inhibition of blank control, kojic acid, TIn, Alk-TIn and VH032-Azi3 + Alk-TIn + SA (0.5 μ M) at 0.1 μ M in A375 (**f**) and B16F10 (**g**) cells, respectively. Data are presented as mean \pm standard deviation (SD). **h, i**, Melanin content of blank control, kojic acid, TIn, Alk-TIn, and VH032-Azi3 + Alk-TIn + SA at 0.1 μ M in A375 (**h**) and B16F10 (**i**) cells, respectively. Data are presented as mean \pm standard error (SE). Statistical analysis was performed using ONE-WAY variance (ANOVA) by Dunnett's multiple comparisons (* $P < 0.05$, **** $P < 0.0001$).

References:

- Yin, X. et al. Chemical Stability of Ascorbic Acid Integrated into Commercial Products: A Review on Bioactivity and Delivery Technology. *Antioxidants* **11**, 153 (2022).
- Weerawardana, M.B.S., Thiripuranathar, G. & Paranagama, P.A. Natural Antibrowning Agents against Polyphenol Oxidase Activity in *Annona muricata* and *Musa acuminata*. *J. Chem.* **2020**, 1904798 (2020).

4. In the in vivo PROTAC anti-tumor experiments, the preformed DeTYR-3 degrader should be included for comparison studies.

Response: Thanks for suggesting this critical control group. We have added the DeTYR-3 group for comparison in related experiments in our revised manuscript and supplementary information. The results are shown in **Fig. R6, Fig. R7 and Fig. R8**. Collectively, the DeTYR-3 group displayed a promising treatment outcome in attenuating the hyperpigmentation, however, the treatment efficacy is not comparable to the VH032-Azi3 + Alk-TIn + SA group.

Fig. R6 (Supplementary Fig. 26 in Supplementary Information). Images of seven different groups of skin creams. SA is added to VH032-Azi3 + Alk-TIn and epi-VH032-Azi3 + Alk-TIn groups.

[editorial note: panel redacted]

Fig. R7 (Fig. 4 in the main text). Therapeutic efficacy evaluation of TYR degradation in a mouse hyperpigmentation model. a, Schematic of the experimental plan in the mouse hyperpigmentation model. **b,** Images of square pigmentation patterns on the backs of mice treated with (1) blank control, (2) VH032, (3) Alk-TIn, (4) VH032 + Alk-TIn, (5) epi-VH032-Azi3 + Alk-TIn + SA, (6) DeTYR 3 and (7) VH032-Azi3 + Alk-TIn + SA creams at different time points (0, 24 and 48 h). **c,** Fontana-Masson (FM) staining of melanin in mouse skin. **d,** Normalized pigmentation level in different treatment groups by digital image analysis using Image J. **e,** Melanin contents of mouse skin samples treated with different groups for 48 h. **f,** Western blot assay of TYR and MITF in mice skin samples treated with different groups for 48 h. **g,** H&E staining of mouse skin tissues treated with different groups for 48 h. Statistical analysis was

performed using ONE-WAY variance (ANOVA) followed by Dunnett's multiple comparisons (**** $P < 0.0001$).

Fig. R8 (Supplementary Fig. 27 in Supplementary Information). Images of the backs of mice treated with control, VH032, Alk-TIn, VH032 + Alk-TIn, epi-VH032-Azi3 + Alk-TIn + SA, DeTYR-3, and VH032-Azi3 + Alk-TIn + SA creams at different time points.

Response to Reviewer #3

In this manuscript, Hu et al. developed endogenous tyrosinase-catalyzed therapeutics for treating melanin-associated skin diseases and drug-resistant tumors. Through the catalytic activity of copper-containing tyrosinase, a series of drug precursors can undergo click reactions in situ to generate effective PROTAC molecules that control disease progression. This is an innovative approach which has been validated to be very effective. Overall, this is an excellent piece of work, and the manuscript is well written. The authors may consider the following comments for revisions.

1. The supporting information do not specify the amount of traditional catalysts, such as copper sulfate/SA. Are their added quantities reasonable, and are they equimolar to mTYR? Beyond fluorescence data, the authors should also provide HPLC evidence to demonstrate that mTYR exhibits superior catalytic efficiency for target molecules compared to traditional catalysts. In Fig. 2, the authors provided the mechanistic studies of TYR's catalytic activity. The advantages of such enzyme-catalyzed reaction and efficiency compared to traditional copper catalysts should be discussed briefly in this section.

Response: Thanks for the insightful comments. We have supplemented the amount of copper sulfate/SA, the HPLC analysis, and the related discussion in our revised manuscript and supplementary information. In this study, the amounts of pS1 and pS2 are 10 μM . The concentration of mushroom tyrosinase (mTYR) is 1 mg/mL. The amounts of CuSO_4 and SA are 8.35 μM and 41.75 μM , respectively. Based on the conversion, the amount of Cu in CuSO_4 is equal to the Cu in mTYR. These new quantitative amounts were added to the figure captions of **Fig. R9**. In addition, we added the quantitative analysis of HPLC analysis for Fluo in **Fig. R10**, which showed a much higher efficiency of catalytic reaction by mTYR than the traditional catalyst CuSO_4 as evidenced by the larger area of detected Fluo in the HPLC. Besides, according to the ab initio molecular dynamics results, the product described a low barrier of 0.32 eV in **Figure 2f**, validating that this is a more facile reaction. Combined with the fluorescence spectra and quantitative analysis of HPLC, TYR significantly enhances the catalytic efficacy compared with traditional CuSO_4 / SA catalysts. Another important benefit of using TYR as the catalyst rather than the traditional copper sulfate/SA is the better biosafety profile. The addition of external Copper could be cytotoxic to the cells; however, leveraging the intracellular TYR as the catalyst could be more biocompatible and with less toxicity.

Fig. R9 (Supplementary Fig. 3 in Supplementary Information). Fluorescence spectra of pS1 + pS2, pS1 + pS2 + CuSO₄ + SA and pS1 + pS2 + mTYR + SA. pS1, pS2 (10 μM); mTYR: mushroom tyrosinase (1 mg/mL); CuSO₄:8.35 μM; SA: 41.75 μM.

Fig. R10 (Supplementary Fig. 4 in Supplementary Information). Quantitative analysis of high performance liquid chromatography analysis of Fluo that is generated in pS1 + pS2, pS1 + pS2 + mTYR, pS1 + pS2 + mTYR + SA, and pS1 + pS2 + CuSO₄ + SA. pS1, pS2 (10 μM), mTYR: mushroom tyrosinase (1mg/mL); CuSO₄:8.35 μM; SA: 41.75 μM. Statistical analysis was performed using ONE-WAY variance (ANOVA) followed by Dunnett's multiple comparisons (*****P* < 0.0001).

2. The confocal imaging data in Fig. 1d show that the fluorescence of “Fluo” is distributed throughout the entire cell, including the nucleus. Why does “Fluo” fluorescence appear in the nucleus considering that based on the data provided in Supplementary Fig. 10, TYR is distributed only in the cytoplasm? The authors should provide a reasonable explanation. Selecting an appropriate imaging time point to achieve colocalization of “Fluo” fluorescence with TYR distribution would make the catalytic role of TYR in generating fluorescence more convincing.

Response: Thanks for catching up on this issue. We have repeated the experiments and supplemented the new data in our revised manuscript and supplementary information. As shown in **Fig. R11** and **Fig. R12**, there is a clear distribution of “Fluo” in the cytoplasm, and further colocalization of “Fluo” with cytosolic TYR.

Fig. R11 (Fig. 1d,e in the main text). Representative confocal microscopy images of A375 (a) and B16F10 (b) cells treated with pS1 + pS2 and pS1 + pS2 + SA, respectively.

Fig. R12 (Supplementary Fig. 7 in Supplementary Information). Representative confocal microscopy images of colocalization of "Fluo" fluorescence with intracellular TYR in A375 and B16F10 cells treated with pS1 + pS2 and pS1 + pS2 + SA, respectively.

3. Will TYR-catalyzed TYR degraders have adverse effects on normal cells? I am curious whether the “Fluo” generation shown in Figure 1d would be higher in the B16F10 cell line compared to normal cell lines. Additionally, the TYR content in skin-related normal cell lines and TYR-overexpressing cell lines (B16F10 and A375) should be quantitatively characterized using flow cytometry or ELISA.

Response: Thanks for the comments. We have supplemented the normal melanocyte cell line to demonstrate the catalytic activity of TYR and characterize the expression of TYR in normal cells in our revised manuscript and supplementary information. **Fig. R13a** barely shows “Fluo” signals in normal melanocytes treated with pS1 + pS2 + SA, which is much weaker compared to those in A375 and B16F10 cells. Thus, we don’t expect any adverse effects on normal cells.

In addition, we tested the expression of TYR in these three cell lines by ELISA. As shown in **Fig. R14**, the results suggest that both A375 and B16F10 cells overexpress TYR, which is about 1.4-fold and 1.8-fold higher than that in normal melanocytes, respectively.

Fig. R13 (Supplementary Fig. 6 in Supplementary Information and Fig. 1d, e in Manuscript Fig 1). Representative confocal microscopy images of normal melanocyte (a), A375 (b), and B16F10 (c) cells treated with pS1 + pS2 and pS1 + pS2 + SA, respectively.

Fig. R14 (Supplementary Fig. 5 in Supplementary Information). Quantitative analysis of TYR expression in normal melanocytes, A375, and B16F10 cells. Statistical analysis was performed using ONE-WAY variance (ANOVA) followed by Dunnett's multiple comparisons (** $P < 0.01$, *** $P < 0.001$).

4. How long can the efficacy of TYR-catalyzed TYR degraders be sustained? While PROTAC molecules can continuously cyclically degrade TYR, this also means the rate of catalytic production of new effective PROTAC molecules will gradually decrease. How to achieve a balance between these two processes? In Fig. 4a, the authors only observed the therapeutic effects over six days. After TYR levels reach a low point, the production of PROTAC molecules may no longer be effective. Could this lead to a recurrence of hyperpigmentation over time? In addition, why does the use of a cream formulation enable effective penetration of TYR degrader precursors into the skin? The authors should use other methods (e.g., fluorescence labeling) to verify that the drug molecules indeed penetrate the skin.

Response: To address the reviewer's concerns, we have supplemented additional western blot (WB) data of catalytic efficacy of DeTYR-3, long-term pigmentation therapeutic results, and the penetration of fluorescence-labeled FAM-VH032-Azi3 and Cy5.5-Alk-TIn in our revised manuscript and supplementary information.

The reviewer raised a great point about the balance of the TYR degradation and PROTAC generation, which is very challenging to achieve. Theoretically, more TYR degradations will induce less PROTAC production, which will weaken the efficacy of PROTACs. However, based on our in vitro and in vivo studies, the powerful TYR degradation capability of DeTYR-3 enabled a very quick attenuation of pigmentation within 48 h. Even though the DeTYR-3 generation is dependent on the intracellular TYR, the catalytic features of DeTYR-3 enabled the relatively long-lasting degradation efficacy, as evidenced by the continuous degradation of TYR in a wash-out experiment (**Fig. R15**). By implementing various drug-incubation times and continuous monitoring time points, we observed the catalytic effects of DeTYRs as evidenced by long-lasting degradation efficacy against TYR.

To address the reviewer's concern about the long-term efficacy of our DeTYR-3, we extended the evaluation timeline on the skin hyperpigmentation mouse model after various treatments. As

shown in **Fig. R1**, we did not see any recurrence of hyperpigmentation over the observation course of 30 days, substantiating the long-term treatment efficacy of in-situ formed DeTYR-3.

Creams, one of the semisolids, are typical drug delivery methods for topical applications, which can passively diffuse through the stratum corneum layer when applied to the skin.^{6,7} Our creams contain mineral oil that can dissolve and carry water-insoluble small molecules across the skin barrier.⁸ To visualize the skin penetration, we modified with fluorescence group FAM to VH032-Azi3 and Cy5.5 to Alk-TIn, respectively. As shown in **Fig. R16**, deeper penetration of the FAM-VH032-Azi3 and Cy5.5-Alk-TIn in the cream group was supported by the fact that the fluorescence group would appear in the epidermis and upper dermis, while it only appeared in the epidermis in the control group.

Fig. R15 (Supplementary Fig. 18 in Supplementary Information). Western blot analysis of continuously degraded TYR in A375 (**a**) and B16F10 (**b**) cells treated with VH032-Azi3 (0.1 μ M) + Alk-TIn (0.1 μ M) + SA (0.5 μ M) at different time points.

Fig. R1 (Supplementary Fig. 28 in Supplementary Information). Images of square pigmentation patterns on the backs of mice treated with blank control and VH032-Azi3 + Alk-TIn + SA creams at different time points.

Fig. R16 (Supplementary Fig. 25 in Supplementary Information). Representative confocal microscopy images of mouse skin tissues treated with FAM-VH032-Azi3 and Cy5.5-Alk-TIn in control and cream groups for 24 h, respectively.

References:

6. Lee, D.H., Lim, S., Kwak, S.S. & Kim, J. Advancements in Skin-Mediated Drug Delivery: Mechanisms, Techniques, and Applications. *Adv. Healthcare Mater.* **13**, 2302375 (2024).
7. Da, D.D. Prodrug Strategies for Enhancing the Percutaneous Absorption of Drugs. *Molecules* **19**, 20780-20807 (2014).
8. Otto, A., Du Plessis, J. & Wiechers, J.W. Formulation effects of topical emulsions on transdermal and dermal delivery. *Int. J. Cosmet. Sci.* **31**, 1-19 (2009).

5. When applied in vivo, particularly for tumor treatment involving the generation of toxic drugs, how can the authors ensure that the click reaction is specifically catalyzed by tumor or disease cells with high TYR expression, rather than by other copper-containing proteins in the blood?

Response: Thanks for your comments. For both applications of in-situ TYR-catalyzed generation of therapeutics, to decrease the potential side effects, the local drug delivery strategies were selected as the administration routes to minimize the systemic exposure. Specifically, the creams are applied to the backs of mice in a specific square pattern for the pigmentation model. Therefore, DeTYR-3 is, in principle, formed in situ by overexpressed TYR in pigmented melanocytes. For the drug-resistant melanoma model, we used peritumoral injections to maximize the exposure of prodrugs to the tumor only. This peritumoral injection for skin tumors offers advantages over a conventional intravenous administration strategy by delivering therapeutics directly around the tumor, offering higher tumor-localized drug concentrations and reduced systemic side effects.^{9, 10}

References:

9. Yan, J. et al. Peritumoral Microgel Reservoir for Long-Term Light-Controlled Triple-Synergistic Treatment of Osteosarcoma with Single Ultra-Low Dose. *Small* **17**, 2100479 (2021).
10. Liu, B. et al. Injectable and NIR-Responsive DNA–Inorganic Hybrid Hydrogels with Outstanding Photothermal Therapy. *Adv. Mater.* **32**, 2004460 (2020).

6. Some sections of the manuscript should include appropriate references to enhance the credibility. For example, the statement: “Recent developments in CuAAC have shifted towards endogenous Cu catalysts, which catalyze chemical reactions without adding exogenous Cu, mitigating the risk of perturbing copper homeostasis and toxicity against normal tissues”.

Response: We have added the related references in our revised manuscript according to the reviewer’s suggestions.

References:

11. Zhu, J. et al. Boosting Endogenous Copper(I) for Biologically Safe and Efficient Bioorthogonal Catalysis via Self-Adaptive Metal-Organic Frameworks. *J. Am. Chem. Soc.* **145**, 1955-1963 (2023).

12. Xue, X. et al. Using bio-orthogonally catalyzed lethality strategy to generate mitochondria-targeting anti-tumor metallodrugs in vitro and in vivo. *Natl. Sci. Rev.* **8** (2020).
13. Liu, Z. et al. Biomarker-activated multifunctional lysosome-targeting chimeras mediated selective degradation of extracellular amyloid fibrils. *Chem* **9**, 2016-2038 (2023).